# Optimizing Temperature for Language Models with Multi-Sample Inference

Weihua Du[1]   Yiming Yang[1]   Sean Welleck[1]

## Abstract

Multi-sample aggregation strategies, such as majority voting and best-of-N sampling, are widely used in contemporary large language models (LLMs) to enhance predictive accuracy across various tasks. A key challenge in this process is temperature selection, which significantly impacts model performance. Existing approaches either rely on a fixed default temperature or require labeled validation data for tuning, which are often scarce and difficult to obtain. This paper addresses the challenge of automatically identifying the (near)-optimal temperature for different LLMs using multi-sample aggregation strategies, without relying on task-specific validation data. We provide a comprehensive analysis of temperature's role in performance optimization, considering variations in model architectures, datasets, task types, model sizes, and predictive accuracy. Furthermore, we propose a novel entropy-based metric for automated temperature optimization, which consistently outperforms fixed-temperature baselines. Additionally, we incorporate a stochastic process model to enhance interpretability, offering deeper insights into the relationship between temperature and model performance. Our code is available at https://github.com/StigLidu/dualdistill.

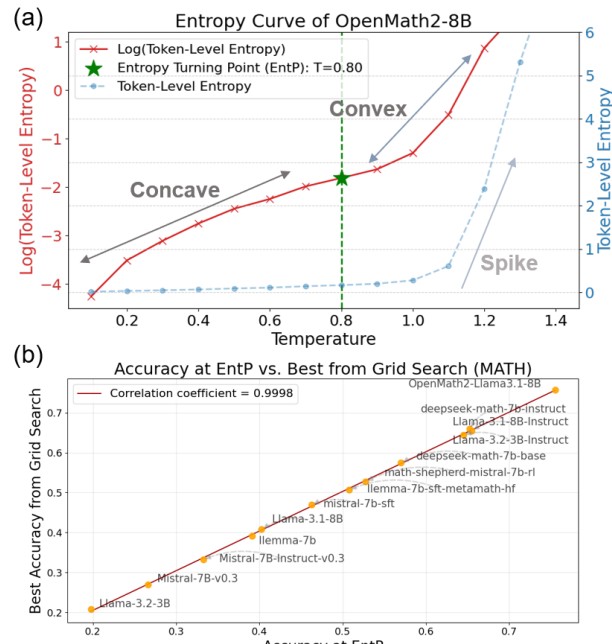

*Figure 1.* **(a) The entropy turning point (EntP)** (green star) is defined as the temperature point where the log-scale of the token-level entropy of generation (red line) shifts from concave to convex, implying the sudden spike in the entropy curve (blue line). **(b)** The accuracy tested at EntP is highly correlated with the best accuracy from grid search over temperatures on the MATH dataset.

## 1. Introduction

Large language models (LLMs) have demonstrated remarkable capabilities across various domains, including question answering (Kamalloo et al., 2023), intelligent agents (Wang et al., 2024b; Zhang et al., 2023), scientific discovery (Ma et al., 2024; Romera-Paredes et al., 2024), and mathematical

[1]Language Technologies Institute, Carnegie Mellon University. Correspondence to: Weihua Du <weihuad@cs.cmu.edu>, Yiming Yang <yiming@cs.cmu.edu>, Sean Welleck <wellecks@cmu.edu>.

*Proceedings of the $42^{nd}$ International Conference on Machine Learning*, Vancouver, Canada. PMLR 267, 2025. Copyright 2025 by the author(s).

reasoning (Ahn et al., 2024; Sun et al., 2024; Lin et al., 2024; Wu et al., 2024). A fundamental research question in generative models is how to effectively sample solutions from a learned distribution and perform inference-time reasoning.

Recently, multi-sample aggregation strategies have gained increasing attention. These strategies involve generating multiple solutions from the underlying distribution and aggregating them into a final prediction (Wei et al., 2022; Yao et al., 2024). Common aggregation techniques, such as majority voting, weighted majority voting, and best-of-N selection, have demonstrated significant performance improvements in benchmark evaluations of LLMs (Welleck et al., 2024; Wang et al., 2024a).

Despite the promising success of multi-sample aggregation strategies, there remains a lack of deep understanding regard-

ing how to optimize the sampling process to enhance LLM performance under different conditions, including variations in training datasets, task types, and model sizes. A crucial open question is how to tune temperature, a key hyperparameter that controls the smoothness of the system-learned distribution. Intuitively, increasing the temperature leads to a smoother distribution, enhancing the diversity of sampled outputs. However, excessively high temperatures can introduce many low-quality samples, making aggregation more challenging (Holtzman et al., 2019; Renze & Guven, 2024). Conversely, lowering the temperature results in a highly concentrated distribution, reducing diversity and potentially omitting high-quality samples. Striking the right balance between over-sampling and under-sampling is therefore essential for optimizing LLM performance.

A common practice in prior evaluations is to use the same temperature across all methods despite variations in training datasets, task types, model sizes, and aggregation strategies. This practice is clearly suboptimal. An alternative approach is to empirically tune the temperature using labeled validation data for each task, dataset, model size, and aggregation strategy (Zhang et al., 2024a; Dhuliawala et al., 2024). However, such a process is tedious and time-consuming and heavily dependent on the availability of labeled validation data, limiting its applicability when such data are scarce.

In this paper, we present the first systematic investigation of how temperature affects LLM performance under multi-sample aggregation strategies across various conditions. Furthermore, we propose a principled algorithmic solution for automated temperature optimization without requiring labeled validation data. Our key idea is as follows:

1. We use the confidence score of each model as a self-assessment measure.

2. If this self-assessment measure is highly correlated with model accuracy on test data, it can serve as a surrogate metric for tuning temperature in the absence of labeled validation data.

A surprising finding from our temperature tuning experiments is the discovery of a phenomenon we term the *entropy turning point (EntP)* in the self-assessed performance curve. As illustrated in Figure 1(a), the token-level entropy (y-axis) of an LLM varies with temperature values (x-axis), shown by the blue curve, while its log-scale representation appears as the red curve. Notably, there is a transition point (EntP) where the red curve shifts from concave to convex. Figure 1(b) shows that the accuracy scores at EntP for a set of LLMs are strongly correlated with their highest accuracy scores obtained through grid-based temperature tuning. This finding supports our intuition that EntP can be leveraged to automatically determine the optimal

temperature for each LLM using multi-sample aggregation strategies. We introduce TURN, our proposed approach for automated temperature optimization. Through extensive experiments, TURN has demonstrated strong generalizability across diverse tasks (*e.g.*, mathematical problem-solving, code generation), model sizes, and aggregation strategies (*e.g.*, majority voting, best-of-N). It consistently outperforms baseline methods using a fixed temperature, yielding significant performance improvements. Additionally, our approach enhances the interpretability of temperature's role in model performance by analyzing EntP. Moreover, our analysis explores how the optimal temperature is influenced by the divergence or similarity between model training and tasks (Section 3).

In summary, TURN provides a novel, efficient, and principled method for optimizing temperature in LLM inference with multi-sample aggregation. It eliminates the need for labeled validation data and significantly improves performance across a wide range of applications.

## 2. Preliminary & Related Work

Before moving to our main contributions, we first review how language models typically generate samples and provide an introduction to multi-sample aggregation strategies.

**Language Model Sampling**  Language models typically generate output for generative tasks by autoregressively sampling from the conditional probability distribution over the next token, given both the input context and previously generated tokens. Formally, for an input sequence $X$ and an output sequence $Y = (y_1, y_2, \ldots, y_N)$, the probability of producing $Y$ is given by the following:

$$P(Y \mid X) = \prod_{i=1}^{N} P(y_i \mid y_{<i}, X). \quad (1)$$

To compute the probability distribution, the model obtains a set of logits $z_i$ and then divides them by a temperature hyperparameter $T$ before applying the softmax function and a regularization function $\mathcal{F}$:

$$P(y_i \mid y_{<i}, X) = \mathcal{F}\left(\text{softmax}\left(\frac{z_i}{T}\right)\right), \quad (2)$$

where $z_i$ is the logit corresponding to token $y_i$. The temperature $T$ controls how peaked or flat the resulting probability distribution will be. The regularization function $\mathcal{F}$ is used to reschedule the sampling process (*e.g.*, Top-$k$ (Kool et al., 2019), Top-$p$ (Holtzman et al., 2019), Min-$p$ (Nguyen et al., 2024) and Locally Typical Sampling (Meister et al., 2023)).

**Multi-Sample Aggregation Strategy**  Since different random seeds can produce varying outcomes, a common approach to mitigate sampling variance is to draw multiple

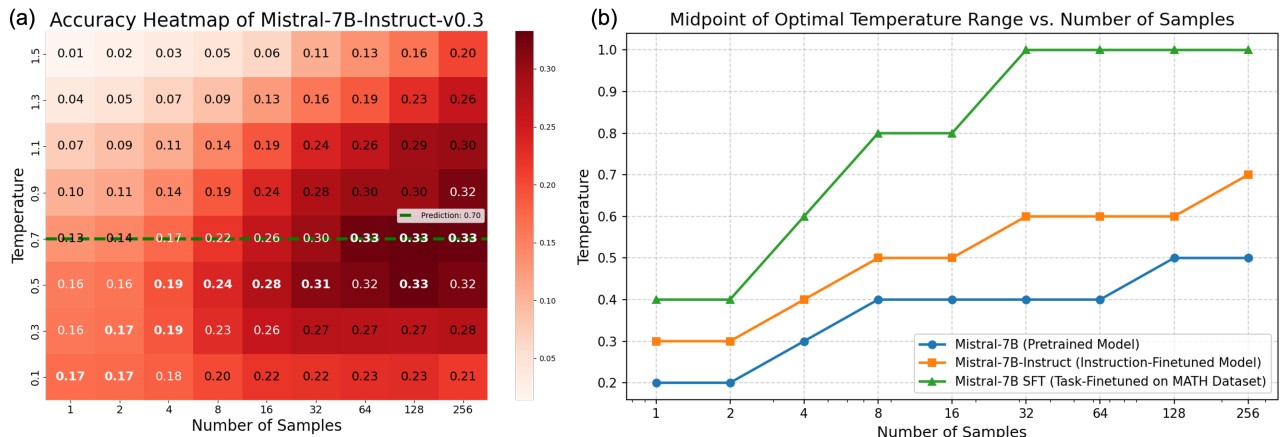

*Figure 2.* **(a) Accuracy Heatmap.** Performance of Mistral-7B-Instruct-v0.3 under majority voting across different temperatures. The best temperature for each sampling size is highlighted in bold white, and the optimal temperature range is shaded white. The green line shows the temperature predicted by our method. **(b) Midpoint of Optimal Temperature Range vs. Number of Samples.** The optimal temperature range varies by model; those with training data more closely matching the task tend to favor higher temperatures.

samples and aggregate their results. In practice, it leads to substantial performance improvements and has been widely adopted to achieve state-of-the-art performance in math reasoning (Sun et al., 2024; Jaech et al., 2024), code generation (Wang et al., 2024a), and many other domains.

Specifically, a set of candidate outputs $Y = \{Y_1, \ldots, Y_N\}$ is generated and then aggregated into a final answer. Two standard aggregation methods are typically employed:

- **Majority Voting**: The final answer is the output that appears most frequently among the candidates, *i.e.*,

$$\hat{y} = \arg\max_{y \in \{Y_1, \ldots, Y_N\}} \sum_{i=1}^{N} \mathbb{I}(Y_i = y),$$

where $\mathbb{I}(\cdot)$ is the indicator function, which returns 1 if its argument is true and 0 otherwise. This method is frequently used where evaluating whether two outputs are equivalent is relatively easy. The method is also called self-consistency (Wang et al., 2022).

- **Best-of-N**: Each sample is scored by a reward function $G$, and the final answer is the one with the highest score:

$$\hat{y} = \arg\max_{y \in \{Y_1, \ldots, Y_N\}} G(y).$$

The reward function $G$ can be defined in various ways, such as a separate language model's likelihood, or a trained or verified reward model.

**Choosing Temperature in Multi-Sample Aggregation** Despite the widespread use of multi-sample aggregation strategies in state-of-the-art systems, the question of choosing the important temperature parameter remains underexplored.

Some studies have investigated selecting a temperature for a single-sample method (Zhang et al., 2024b; Li et al., 2024; Kumar & Sarawagi, 2019; Xie et al., 2024; Dhuliawala et al., 2024) or multi-sample aggregation with a validation set (Zhang et al., 2024a). Our method has two key differences: (1) we focus on state-of-the-art multisample aggregation strategies rather than single-sample inference, and (2) we find the optimal temperature without validation data.

## 3. Correlation Between Model Training and Optimal Temperature

Multi-sample aggregation strategies—commonly used in problem-solving, code generation, and related domains—leverage information from multiple samples, which helps escape local minima and improve robustness. In these settings, *sample diversity* becomes crucial: a diverse set of candidate samples increases the likelihood that the correct solution appears in the pool, rather than repeating the same mistake. The *temperature* parameter is a primary lever for controlling this diversity.

We hypothesize that how a model is trained impacts the optimal temperature for multi-sample inference strategies. In particular, a more specialized or fine-tuned model can safely explore higher temperatures without drifting into low-quality outputs. In contrast, a general-purpose model typically benefits from a lower temperature to remain focused on relevant content.

We investigate this in two steps: In Section 3.1, we show that the optimal temperature varies for a base, instruction-tuned, and fine-tuned model. Then in Section 3.2, we establish a general relationship between a model's proximity to the

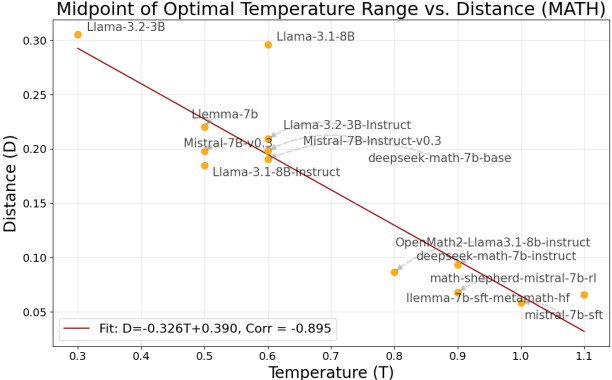

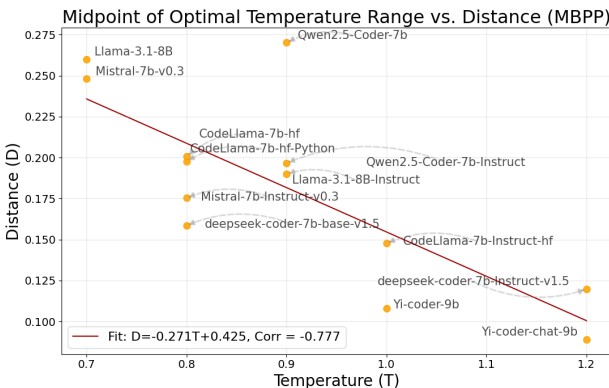

*Figure 3.* Plot of midpoints of optimal temperature ranges (x-axis, sample size 128) vs. distances between models and tasks (y-axis). A strong negative correlation is observed on the MATH and MBPP datasets, with correlation coefficients of -0.895 and -0.777.

target task and its corresponding optimal temperature. Our key insight is that token-level entropy is a proxy of distance from a task, which motivates our entropy-based method for automatic temperature selection in Section 4.

### 3.1. Optimal Temperature Range Varies

We first demonstrate that the optimal sampling temperature varies by model type. We test three *Mistral-7B* variants: the *pretrained base model*, the *instruction-finetuned version (Mistral-7B-Instruct)*, and a *task-finetuned model for MATH*[1] (Wang et al., 2024c). Each model is evaluated using multi-sample aggregation across different temperatures. Figure 2(a) presents the accuracy heatmap for the Mistral-7B-Instruct model on the MATH dataset. At smaller sample sizes, lower temperatures tend to produce better accuracy. However, higher temperatures can yield better results as the sample size increases. For a fixed sample size, the accuracy curve follows a single-peak pattern: it rises as temperature increases and peaks, and then gradually declines, staying relatively steady near the peak.

Since the single-peak behavior, we define the $\epsilon$-**optimal**

[1]Model link: https://huggingface.co/peiyi9979/mistral-7b-sft

**temperature range**. This range encompasses temperatures $T$ where the accuracy $A(T)$ is no less than $A(T^*) - \epsilon$, with $A(T^*)$ representing the peak accuracy. Given the curve's single-peak nature, this range forms an interval around $T^*$. For our analysis, we set $\epsilon = 0.02$, effectively capturing the temperatures close to the peak where the accuracy remains relatively high.

We then plot the midpoint of this optimal temperature range for each model variant and various sample sizes (Figure 2(b)). We observe that the pretrained model has the lowest midpoint, the instruction-finetuned model has a higher midpoint, and the task-finetuned model has the highest. Another observation is that optimal temperature ranges change slowly once beyond a sample size of 32. Therefore, we choose a sample size of 128 in our following experiments to ensure stable performance in the rich-sample setting.

From these observations, we hypothesize a general relationship between how closely a model is tuned to a particular task and the temperature that yields the best accuracy. We discuss this hypothesis further in the next section.

### 3.2. Correlation Between Training-Task Similarity and Optimal Temperature

Our goal is to establish a general relationship between a model's learned distribution and its optimal temperature for a task. Our key intuition is that token-level entropy can serve as a proxy for how distant a model is from a target task and that this distance helps identify the optimal temperature.

Specifically, we define a distance metric that measures how similar a model's training data is to a given task. Let $\mathcal{T} = \{X_1, ..., X_k\}$ be the task with $k$ problem instances. We define this distance $\mathcal{D}(\mathcal{M}, \mathcal{T})$ as the average of token-level entropy $\mathcal{H}(.)$ of the language model $\mathcal{M}$ when generating the answers $\mathcal{A} = \{Y_1, ..., Y_k\}$ for the problems in $\mathcal{T}$:

$$\mathcal{D}(\mathcal{M}, \mathcal{T}) = \frac{1}{k} \sum_{i=1}^{k} \left[ \frac{1}{|Y_i|} \sum_{j=1}^{|Y_i|} \mathcal{H}\big(p_{\mathcal{M}}\big(\cdot \mid X_i, Y_{i,<j}\big)\big) \right],$$
(3)

where
$$\mathcal{H}(p) = -\sum_{v \in p} p(v) \log p(v).$$
(4)

To avoid bias toward ground-truth references, we use model-generated sequences $\{Y_i\}$ instead of official gold solutions. Meanwhile, the distance is measured at a low temperature $T = 0.5$ to ensure the generation stability.

We evaluated several language models on the MATH and MBPP datasets, including pretrained, instruction-finetuned, and task-finetuned models. Figure 3 plots the midpoint of the optimal temperature range against our distance metric, demonstrating a strong negative correlation. Specifically,

across our model set, the correlation on MATH is $-0.895$, while on MBPP it is $-0.777$.

In practice, this suggests using a higher temperature (*e.g.*, $T = 0.9$–$1.1$) for task-finetuned models and a lower temperature (*e.g.*, $T = 0.5$–$0.7$) for more general-purpose models (pretrained or instruction-finetuned).

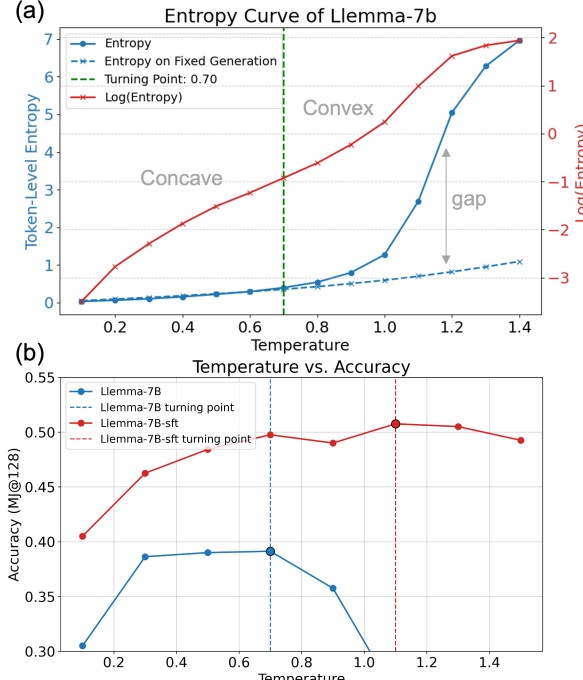

*Figure 4.* **Entropy Curve Characteristics.** **(a)** The token-level entropy $\mathcal{H}$ (solid blue line) increases slowly at lower temperatures and then jumps sharply at a critical turning point. In contrast, the entropy for a fixed (greedy) generation stays low (dotted blue line). $\log(\mathcal{H})$ (red line) reveals a transition from concavity to convexity that aligns with the sharp increase in $\mathcal{H}$, marking the entropy turning point (EntP). **(b)** EntP aligns with the best temperature and varies across different models.

# 4. Entropy-Based Automatic Temperature Selection

Determining an optimal sampling temperature is crucial in multi-sample aggregation strategies, yet existing approaches often rely on labeled data or tuning on a validation set. This reliance becomes problematic when no such data are available. In this section, we show how to leverage token-level entropy as an intrinsic property to pinpoint a suitable temperature without labeled data. We first demonstrate a spike on *token-level entropy* as a signal of quality collapse in Section 4.1. Then develop a method that automatically selects temperature using an *entropy turning point (EntP)* derived from the spike in Section 4.2. Finally, we applied a stochastic process model to explain the mechanism of our

algorithm in Section 4.3.

## 4.1. Entropy Spike as an Indicator of Quality Collapse

First, we discover a surprising phenomenon that we call the entropy spike. Specifically, increasing the temperature smoothly increases the model's entropy, until a dramatic spike where the entropy rapidly increases. We believe the spike is a good indicator of sample quality collapse.

As illustrated in Figure 4(a), we calculate the token-level entropy at different temperature levels (solid blue line). To reduce computational overhead, we compute the entropy only over the top-$K$ tokens (with the highest probabilities) at each step, setting $K = 1000$ in all subsequent experiments. The entropy curve remains stable for lower temperatures but then shows a sudden rise. One might attribute this behavior to temperature's role in flattening the distribution (Equation 1). However, the following analysis indicates that this spike reflects a substantial change in the model's next-token distribution.

Specifically, we constrain the model to generate the same outputs produced by greedy decoding while evaluating entropy under a higher temperature (dotted blue line). If temperature alone were responsible for the entropy spike, these fixed outputs would yield a similarly high entropy. However, as shown in Figure 4(a), we observe a significant gap between these two entropy curves, indicating that the actual sampling distribution undergoes a large shift.

Thus, we infer that the sudden rise in the entropy curve implies a substantial drop in sample quality. Setting the temperature around this sudden rise can balance sufficient diversity without a large quality drop, which is suitable for multi-sample aggregation strategies.

## 4.2. Turning Point Temperature Selection (TURN)

Given the token-level entropy curve of a language model on a specific task, how can we identify a suitable temperature for multi-sample aggregation strategies? Inspired by the difference in the shapes of the entropy curve: When the temperature remains low, the entropy increases *flatly*. However, when the sampling temperature is near the spike, the entropy increases *(super)-exponentially*, implying a quality drop in samples. Therefore, after taking the logarithm of the entropy curve (shown in Figure 4(a), red line), the flat part becomes concave while the exponentially-increasing part becomes convex. We define the *entropy turning point (EntP)* as the temperature where the log entropy curve becomes convex. Figure 4(b) tests the llemma-7b base model and its task-finetuned variant[2] (Sun et al., 2024), and EntP matches the position with the highest accuracy and varies between dif-

---

[2] Model link: https://huggingface.co/ScalableMath/llemma-7b-sft-metamath-level-1to3-hf

ferent models. Based on EntP, we develop a new method for automatic temperature prediction in multi-sample aggregation strategies, called Turning Point Temperature Selection (TURN).

The optimal temperature should be around EntP to achieve both sample quality and diversity. At the same time, we found that some aggregation methods may be more tolerant of quality drops (*e.g.*, for best-of-N, only one sample is enough to be correct). So we added a small adaptation factor $\beta$ based on the aggregation function, and it is set to 0 and +0.1 for majority voting and best-of-N, respectively. The aggregation adaptation for best-of-N is calculated in the MATH dataset but can be directly applied to other tasks. Refer to Appendix C for more details.

Specifically, given a language model $\mathcal{M}$, a task $\mathcal{T} = \{X_1, \ldots, X_k\}$ with $k$ input instances, and an aggregation method $\mathcal{A}$. To estimate token-level entropy, we randomly sampled $N$ times. In each time, we randomly choose an input instance $X_i$ and generate one sample by $\mathcal{M}$ under each candidate temperature $t_j = j \cdot t$ (with $t$ being the temperature interval and $j = 0, 1, \ldots, J$, where $J = \lfloor t_{\max}/t \rfloor$). These entropies are then aggregated to calculate the average entropy $\mathcal{H}(t_j)$ at each temperature $t_j$. Taking the logarithm, we obtain $\ell(t_j) = \log \mathcal{H}(t_j)$.

Next, we identify the EntP index $j^*$, where the second derivative of $\ell$ changes from negative to positive, and select its corresponding temperature $j^* \cdot t$. Then we add the aggregation adaptation factor $\beta$ to form the final prediction. The pseudocode for our algorithm is listed in Algo. 1.

---

**Algorithm 1** Turning Point Temperature Selection (TURN)

---

1: **Input:** Language Model $\mathcal{M}$, task $\mathcal{T} = (X_1, ..., X_k)$, temperature interval $t$, maximum temperature $t_{\max}$, sample size $N$, aggregation method $\mathcal{A}$.
2: **Output:** Predicted Temperature $T_{\text{pred}}$.
3: Compute $J = \lfloor t_{\max}/t \rfloor$ {Number of choices}
4: Initialize entropy list $\mathcal{E} = []$
5: **for** $n = 1$ to $N$ **do**
6:     Randomly select $X_i$ from $\mathcal{T}$
7:     **for** $j = 0$ to $J$ **do**
8:         Generate a sample $Y$ using $\mathcal{M}$ with $T = j \cdot t$
9:         Compute token-level entropy of $Y$, add to $\mathcal{E}[j]$
10:     **end for**
11: **end for**
12: Compute $\mathcal{H}(j) = \text{Mean}\,(\mathcal{E}(j))$ for all $j$
13: Compute $\ell(j) = \log \mathcal{H}(j)$ for all $j$
14: Find $j^* = \arg\min_j \left( \frac{d^2\ell}{dt^2} > 0 \right)$
15: Compute $t^* = j^* \cdot t$
16: Add adaptation factor $\beta_{\mathcal{A}}$: $T_{\text{pred}} = t^* + \beta_{\mathcal{A}}$
17: **Return** $T_{\text{pred}}$

---

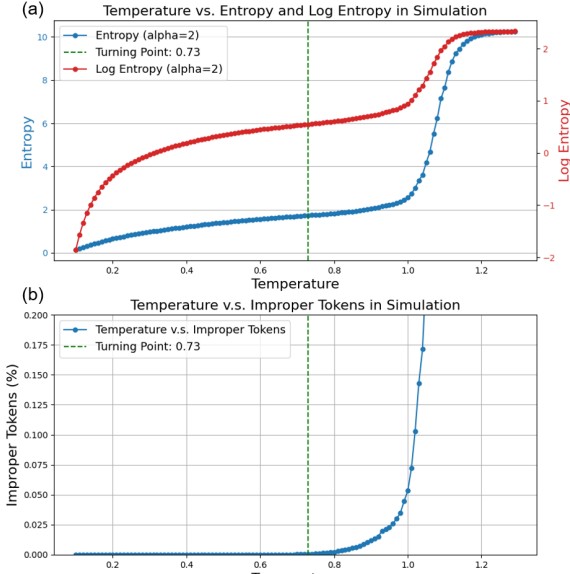

*Figure 5.* **Stochastic Process Model.** We run our process model in the setting: $N_0 = 10$, $N_1 = 30000$, $L_0 = 0$, $\sigma_0 = 1$, $L_1 = -10$, and $\alpha = 2$. **(a)** The entropy curve is similar to that of the real language model: flat at first, and then sharply increases. **(b)** We calculate the relation between temperature and the percentage of improper tokens in the simulation, and the percentage of improper tokens quickly increases after EntP.

### 4.3. A Stochastic Process Model

We applied a stochastic process model to explain why the entropy curve exhibits a sudden spike and what that spike signifies.

Because inference is sequential, when the language model makes an error (for example, by sampling an improper token), it increases the likelihood of further mistakes. Meanwhile, the model may occasionally recover and return to a correct trajectory.

To simulate this process, we adopt a stochastic process model with $K$ steps in sequential, generating a token in each step. At the start, the model has an initial error rate $p = p_{\text{init}}$, representing the probability of selecting an improper token. At each step, if the model selects an improper token, the likelihood of further errors increases to $1 - (1 - p)^\alpha$, where $\alpha > 1$ is called the noise tolerance rate. Conversely, if the model selects a proper token, the error probability decreases to $p^\alpha$ (but cannot be smaller than $p_{\text{init}}$).

To build a bridge between the temperature $T$ and the initial error rate $p_{\text{init}}$, we propose an estimation. All tokens are labeled proper or improper irrelevant to contexts, and the number of improper tokens ($N_1$) is much larger than that of proper tokens ($N_0$). In the beginning, proper tokens have high logits $L_0$ with a variance $\mathcal{N}(0, \sigma_0^2)$ to reflect

the nature that there may be several proper next tokens with similar logits. Improper tokens have uniformly low logits $L_1$. Then, the initial error rate $p_{init}$ is determined as the probability of selecting an improper token based on the logits and temperature. During inference, all improper tokens equally share the error rate $p$, while proper tokens account for the remaining probability based on their logits.

Using this setup, we can estimate the token-level entropy. As shown in Figure 5(a), the simulated entropy curve (blue line) aligns well with the observed entropy curves of a real language model (Figure 4(1) solid blue line). Meanwhile, Figure 5(b) shows the relationship between the temperature and the percentage of improper tokens, which rises quickly after EntP. This observation suggests that, before EntP, increasing the temperature can help explore the proper tokens. However, after EntP, the increase in the percentage of improper tokens makes the model uncertain and creates errors, implying a quick drop in sample quality. The behavior of the stochastic process model is consistent with our observations of language models, proving that token-level entropy is a good indicator of sample quality. Detailed formulas and experiments can be found in Appendix B.

## 5. Evaluating TURN

We want to answer the following research questions about our approach TURN for selecting the optimal temperature:

- **RQ1**: How is the accuracy of TURN in automatic temperature prediction?

- **RQ2**: How efficient is TURN regarding the number of samples (the parameter $N$ in Algo. 1)?

Through experiments, TURN proves effective across models, aggregation strategies, and tasks while remaining efficient, requiring only a few samples for temperature prediction.

### 5.1. Experiment Setup

We evaluate our methods in two scenarios where sampling-based inference is widely used: *Math Problem Solving with Majority Voting* and *Code Generation with Best-of-N*. The datasets and models are as follows:

**Math Problem Solving:** We assess language models' reasoning abilities using the MATH dataset (Hendrycks et al., 2021), which consists of competition-level math problems. To accommodate multiple models, we randomly select 200 test problems (40 per difficulty level). Accuracy is measured based on majority voting. We test general-purpose models (Llama (Dubey et al., 2024), Mistral (Jiang et al., 2023)), domain-specific models (Llemma (Azerbayev et al., 2023), OpenMath2 (Toshniwal et al., 2024), Deepseek-Math (Shao

*Table 1.* The prediction from our algorithm TURN *(Pred.)*, the *optimal temperature ranges (Opt. Range)* from grid search, and the *performance drop (PD)* for various models tested in the MATH and MBPP datasets. TURN achieved hit rates of $12/13$ and $11/13$, average temperature gaps of $0.023$ and $0.015$, and average performance drop of $0.32\%$ and $0.59\%$.

| MATH with Majority Voting (Hit Rate: 92.3%) | | | |
|---|---|---|---|
| **Model Name** | **Pred.** | **Opt. Range** | **PD ($\downarrow$)** |
| mistral-7b-sft | **0.9** | 0.5–1.5 | 0.75% |
| math-shepherd-mistral-7b-rl | **0.9** | 0.5–1.3 | 0% |
| Mistral-7B-v0.3 | **0.7** | 0.3–0.7 | 0.37% |
| Mistral-7B-Instruct-v0.3 | **0.7** | 0.5–0.7 | 0% |
| deepseek-math-7b-base | **0.6** | 0.5–0.7 | 0.5% |
| deepseek-math-7b-instruct | **0.8** | 0.5–1.3 | 0.75% |
| llemma-7b | **0.7** | 0.3–0.7 | 0% |
| llemma-7b-sft-metamath-hf | **1.1** | 0.7–1.5 | 0% |
| Llama-3.1-8B-Instruct | **0.6** | 0.3–0.7 | 0% |
| Llama-3.1-8B | **0.6** | 0.5–0.7 | 0.5% |
| Llama-3.2-3B-Instruct | **0.7** | 0.5–0.7 | 0% |
| Llama-3.2-3B | 0.6 | 0.3 | 1% |
| OpenMath2-Llama3.1-8B | **0.8** | 0.5–1.1 | 0.25% |

| MBPP with Best-of-N (Hit Rate: 84.6%) | | | |
|---|---|---|---|
| **Model Name** | **Pred.** | **Opt. Range** | **PD ($\downarrow$)** |
| deepseek-coder-7b-base-v1.5 | 1.0 | 0.7–0.9 | 2.70% |
| deepseek-coder-7b-instruct-v1.5 | 1.0 | 1.1–1.3 | 1.77% |
| CodeLlama-7b-hf | **0.8** | 0.7–0.9 | 0% |
| CodeLlama-7b-Python-hf | **0.9** | 0.7–0.9 | 0.71% |
| CodeLlama-7b-instruct-hf | **0.9** | 0.9–1.1 | 0% |
| Qwen2.5-Coder-7B | **0.9** | 0.7–1.1 | 0% |
| Qwen2.5-Coder-7B-Instruct | **0.9** | 0.7–1.1 | 0% |
| Yi-coder-9B | **0.7** | 0.7–1.3 | 0.97% |
| Yi-Coder-9B-chat | **0.9** | 0.9–1.5 | 0% |
| Llama-3.1-8B | **0.9** | 0.5–0.9 | 1.08% |
| Llama-3.1-8B-Instruct | **0.8** | 0.7–1.1 | 0.39% |
| Mistral-7B-v0.3 | **0.8** | 0.5–0.9 | 0% |
| Mistral-7B-Instruct-v0.3 | **0.7** | 0.7–0.9 | 0% |

et al., 2024)), and fine-tuned models (Math-Shepherd (Wang et al., 2024c), Easy-to-Hard (Sun et al., 2024)).

**Code Generation:** For code generation, we use the MBPP dataset (Austin et al., 2021), selecting the first 100 programming problems. Accuracy is measured using pass@K, where correctness is determined by passing provided unit tests. We regard the unit tests as the best-of-N strategy with a perfect reward model to rank answers. Besides general-purpose models, we evaluate code-specific models, including Deepseek-Coder (Guo et al., 2024), CodeLlama (Roziere et al., 2023), Qwen2.5-Coder (Hui et al., 2024), and Yi-coder (01.AI, 2024).

**Implement Details:** For both tasks, we sample 256 times

*Table 2.* **Comparison Between TURN and Fixed Temperatures.** We compared TURN to various fixed temperatures under two metrics: The sum of *Temperature Gap* and the average *Performance Drop*. '-Ada.' means removing the aggregation adaptation factor $\beta$. Although some temperatures are generally suitable for multi-sample aggregation strategies (*i.e.*, $T = 0.7$ or $T = 0.9$), TURN can outperform any single fixed temperature across any dataset, highlighting the strong performance of TURN in automatic temperature selection. The underline means not inferior to the best fixed temperature, and the bold is the best result.

| | | Fixed Temperature | | | | | | TURN | TURN |
|---|---|---|---|---|---|---|---|---|---|
| | | 0.1 | 0.3 | 0.5 | 0.7 | 0.9 | 1.1 | | -Ada. |
| **MATH** | Sum Gap ($\downarrow$) | 4.6 | 2.0 | _0.4_ | _0.4_ | 2.0 | 3.6 | **_0.3_** | **_0.3_** |
| | Avg. Drop ($\downarrow$) | 8.6% | 3.5% | 1.0% | _0.8%_ | 3.1% | 7.2% | **_0.3%_** | **_0.3%_** |
| **MBPP** | Sum Gap ($\downarrow$) | 8.2 | 5.6 | 3.0 | 0.8 | **_0.2_** | 1.2 | **_0.2_** | 0.6 |
| | Avg. Drop ($\downarrow$) | 22.5% | 10.7% | 5.1% | 1.5% | 0.9% | 4.2% | **_0.5%_** | 1.1% |
| **Average** | Sum Gap ($\downarrow$) | 6.4 | 3.8 | 1.7 | _0.6_ | 1.1 | 2.4 | **_0.25_** | _0.45_ |
| | Avg. Drop ($\downarrow$) | 15.55% | 7.1% | 3.05% | _1.15%_ | 2.0% | 5.7% | **_0.4%_** | _0.7%_ |

per question at each temperature level and compute accuracy across different sampling sizes. For temperature prediction in TURN, we use an interval of $t = 0.1$ and set $N = 8 \times$ dataset size (an excessive sample size, see Section 5.4 for discussion). Additional inference configurations are detailed in Appendix A.

*Table 3.* **Variance of Entropy Estimation.** We report the average variance of the entropy curve and the variance of estimated temperature $T_{\text{pred}}$ under different sample sizes with 50 trials on Llama3.1-8B-Instruct on MATH. A small sample size (*e.g.*, 40) is sufficient for entropy estimation in TURN for its low prediction variance and small performance drop.

| | Sample Size ($N$) | | | |
|---|---|---|---|---|
| | 10 | 40 | 100 | 800 |
| Mean (Var ($\mathcal{H}(.)$)) | 0.084 | 0.022 | 0.010 | 0.001 |
| Var ($T_{\text{pred}}$) | 0.020 | 0.005 | 0.003 | 0.001 |
| Performance Drop($\downarrow$) | 0.9% | 0.2% | 0.1% | 0.0% |

### 5.2. Evaluation Metrics

To assess the performance of our algorithm for automatically selecting the optimal sampling temperature, we define the following key metrics (all the metrics are calculated under a large sample size of 128, refer to Section 3.1 for discussion):

**Metrics:** We use the following metrics to evaluate the accuracy and reliability of our temperature prediction algorithm:

- **Hit Rate (HR):** The frequency with which TURN selects a temperature within the $\epsilon$-*optimal range*[3], indicating practical reliability.

- **Temperature Gap (TG):** The absolute difference between the predicted temperature and the nearest boundary of the $\epsilon$-*optimal temperature range*.

- **Performance Drop (PD):** The accuracy loss compared to the best temperature found by grid search.

---

[3]Defined in Section 3.1.

### 5.3. Baseline

As no existing method automatically adjusts temperatures in multi-sample aggregation strategies, we compare against a **fixed temperature** baseline. We search for $\{0.1, 0.3, 0.5, 0.7, 0.9, 1.1\}$ and select the temperature that maximizes the overall accuracy. This mimics a common, yet suboptimal practice where developers apply a single temperature across all models, disregarding variations in model behavior and task requirements.

### 5.4. Results

We evaluated 13 models on two tasks—MATH (with majority voting) and MBPP (with Best-of-N)—and present the results in Table 1. Recall Figure 1(b), the *correlation coefficient* between the accuracy of the predicted temperature and the best accuracy from grid search is 0.9998 for MATH (and 0.9913 for MBPP). TURN achieves a Hit Rate of $12/13$ on MATH and $11/13$ on MBPP, indicating strong performance across most models. The Temperature Gap remains minimal even when the predicted temperature falls outside the $\epsilon$-optimal range (0.023 for MATH and 0.015 for MBPP). Compared to the best temperatures found through the grid search, TURN incurs only a small drop in average performance (0.32% and 0.59%, respectively). Full per-model results and predicted turning points are provided in Appendix D.

**Comparison with Fixed Temperatures:** We next compare TURN to a fixed temperature baseline. Specifically, we sample temperatures from 0.1 to 1.1 at intervals of 0.2 and report the *Temperature Gap (TG)* and *Performance Drop (PD)* in Table 2. Our method outperforms the best of fixed temperatures by 0.5% on MATH and 0.4% on MBPP in average accuracy. When both tasks are combined, the margin increases to 0.75%, highlighting the benefit of adaptive temperature selection over a uniform fixed temperature.

**Number of Samples for Temperature Estimation:** Fi-

nally, we assess the efficiency of TURN by examining the prediction variance under different sample sizes for Llama-3.1-8B-Instruct on MATH. As shown in Table 3, we report the average variance of the entropy curve in all choices of $T$, the variance of the predicted temperature, and the average performance drop. We find that even with a moderate sample size (*e.g.*, $40$ samples), the variance remains low and the performance drop is tiny (0.2%), suggesting that a small sampling budget is sufficient for accurate temperature estimation and thus proves the efficiency of our algorithm.

## 6. Conclusion

In this paper, we investigated the critical role of temperature in multi-sample aggregation strategies. We observed that the optimal temperature varies significantly across models due to differences in training strategies and data distributions. By analyzing the relationship between training-testing distribution similarity and the optimal temperature range, we identify a strong correlation that provides valuable insights into model behavior. Furthermore, we proposed the first method for automatically predicting optimal temperatures across diverse tasks, achieving this without labeled data. Our findings contribute to a deeper understanding of temperature's impact on language model performance and offer a practical approach for optimizing inference settings.

## Impact Statement

This paper presents work whose goal is to advance the field of Machine Learning. There are many potential societal consequences of our work, none which we feel must be specifically highlighted here.

## Acknowledgments

SW would like to thank Convergent Research and the Lean Focused Research Organization and the Microsoft Accelerating Foundation Models Research Initiative.

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

# A. Inference Configuration

## A.1. Software

Our experiments build upon two open-source projects: *Easy-to-Hard Generalization* (Sun et al., 2024) for the MATH dataset and *bigcode-evaluation-harness* (Ben Allal et al., 2022) for the MBPP dataset. We employ *vLLM* (Kwon et al., 2023) to accelerate inference. All experiments can be reproduced on a single L40S or A6000 GPU.

## A.2. Sampling Hyperparameters

We use zero-shot inference for models fine-tuned specifically for each dataset. For general-purpose models, we use four in-context examples (few-shot inference) to ensure correct output formatting. The maximum output length is set to 1024 tokens for all tasks. For the MATH dataset, we use top-k sampling with $k = 20$. No additional sampling constraints are imposed for the MBPP dataset.

## A.3. Metric Calculation

To compute the majority vote results for the MATH dataset, we consider two samples to have the same answer if they match after normalization. For the pass@K metric, we follow the definition in Chen et al. (2021). Let $N$ be the total number of samples and $C$ be the number of correct samples. Then $\text{pass@}K$ is defined as:

$$\text{pass@}K = 1 - \frac{\binom{N-C}{K}}{\binom{N}{K}}. \tag{5}$$

# B. Details of the Stochastic Process Model

We introduce a stochastic process model to explain that (1) the token-level entropy increases steadily at the beginning but rises rapidly when the sampling temperature reaches a certain threshold. (2) The optimal temperature is near the turning point when using multi-sample aggregation strategies.

The stochastic process model has two underlying assumptions: (1) Every token can be labeled as 'proper' or 'improper' at each decoding step. Generally, proper tokens have relatively higher logits than improper tokens, while the number of improper tokens is much higher than that of proper tokens. (2) When an improper token is generated, improper tokens have a higher generation probability in the next step, and vice versa.

Under these two assumptions, we can calculate the token-level entropy under different sampling temperatures, and the temperature-entropy curve fits that of real language models. Meanwhile, the percentage of improper tokens quickly increases after the turning point, implying a quick drop in sample quality in real language models.

## B.1. Model Setup

### B.1.1. INITIAL CONDITIONS

We consider a discrete-time process $\{x_t\}_{t=0}^{K}$ where each $x_t \in [0, 1]$ represents the model's probability of producing an improper token at time step $t$. We start with an initial error rate:

$$x_0 = x_{\text{init}} \in [0, 1].$$

Conceptually, $x_{\text{init}}$ corresponds to the model's baseline 'error propensity' at the start. This value is related to the sampling temperature $T$ of the language model: higher $T$ typically yields a flatter probability distribution over tokens, increasing the chance of selecting an improper token and thus increasing $x_{\text{init}}$. (See Section B.1.4 for a heuristic link between temperature and initial error rate.)

### B.1.2. INTERPRETING THE ERROR RATE

At each step $t$, the language model chooses a single token, and each token is classified as proper or improper. Although in practice, the correctness of a token depends on the context and is not truly binary, we approximate this by treating correctness as a Bernoulli trial:

- Probability of producing an improper token: $x_t$;

- Probability of producing a proper token: $1 - x_t$.

Define a random variable $E_t$ that indicates whether an error occurred at time step $t$:

$$E_t = \begin{cases} 1 & \text{with probability } x_t, \text{ (improper token)}, \\ 0 & \text{with probability } 1 - x_t, \text{ (proper token)}. \end{cases}$$

### B.1.3. ERROR RATE UPDATE RULES

After each step, the error rate of the next step $x_{t+1}$ is updated based on whether the token in this step $t$ is proper or not:

**If an error occurs** ($E_t = 1$): The error rate of the next step $x_{t+1}$ is increased. Intuitively, making a mistake can make the model more likely to continue making errors. Formally, we update:

$$x_{t+1} = 1 - (1 - x_t)^\alpha.$$

It can be seen that $x_{t+1} \geq x_t$ for $x_t \in [0, 1]$ and $\alpha > 1$ ($\alpha$ is a hyperparameter). Here, $\alpha$ can be considered as the noise tolerance rate, measuring how stable the model is when it experiences unexpected noise, and we try different $\alpha$ in experiments.

**If a proper token is produced** ($E_t = 0$): The error rate of the next step $x_{t+1}$ is reduced, reflecting a 'reinforcement' of correct behavior. We do this by:

$$x_{t+1} = \max(x_t^\alpha, \; x_{\text{init}}).$$

It generally makes $x_{t+1} \leq x_t$ smaller, so this update lowers the error rate. In particular, we do not allow the error rate to drop below the initial baseline $x_{\text{init}}$.

### B.1.4. LINKING INITIAL ERROR RATE AND TEMPERATURE

At time step $t$, the token probability mass of the model is divided into:

- **Improper tokens** with total probability $P_{1,\text{improper}} = x_t$;

- **Proper tokens** with total probability $P_{0,\text{proper}} = 1 - x_t$.

Therefore, by definition, we have $x_{\text{init}} = P_{1,\text{improper}}$. Under higher temperatures $T$, the softmax distribution flattens, increasing $P_{1,\text{improper}}$ because the number of improper tokens is large but their logits are low. Thus, $x_{\text{init}}$ increases as $T$ increases.

### B.1.5. TYPE OF TOKENS DURING DECODING

The probability of tokens when decoding is usually multi-peak (*i.e.*, except for the token with the highest logit, some other tokens have reasonably high logits and are also acceptable during decoding), so it is natural to consider a scenario with three categories of tokens:

- **Proper tokens:** Small number of tokens with high logits. Let $N_0$ be the number of proper tokens. To capture the multi-peak behavior, the logits of proper tokens $l_{0,1}, ..., l_{0,N_0}$ are sampled from the Gaussian distribution $\mathcal{N}(L_0, \sigma_0)$.

- **Low Probability Improper tokens:** Many low-logit tokens where language models seldom choose them. Let $N_1$ be the number of tokens of this type, and their logits are set to $L_1$ for simplicity.

- **High Probability Improper tokens:** Due to insufficient training or mistakes in training data, some tokens may have exceptionally high logits but are logically improper in (*e.g.*, the token 3 in $1 + 1 = 3$). Since different language models behave differently, we only consider decoding without high-probability improper tokens in our discussion.

For the first two types of tokens, we have $L_0 > L_1$, $N_0 \ll N_1$.

### B.1.6. TOKEN PROBABILITY DURING SAMPLING

At each step $t$, the probability of producing improper tokens is $x_t$. Let $p_{t,\text{proper/improper},j}$ be the probability of the $j$-th proper/improper tokens at step $t$. For improper tokens, we have:

$$p_{t,\text{improper},j} = \frac{x_t}{N_1}, \quad \forall j \in [1, N_1].$$

For the proper tokens, we allocate the remaining probability $1 - x_t$ according to their relative logits:

$$p_{t,\text{proper},j} = (1 - x_t)\text{softmax}(l_{0,1}, ..., l_{0,N_0})_j, \quad \forall j \in [1, N_0].$$

This ensures that the relative order of the probabilities for the proper tokens remains determined by their logits, while the total mass allocated to the improper tokens is $x_t$.

### B.1.7. ENTROPY CALCULATION

We define token-level entropy $\mathcal{H}$ on a sequence of decoding steps:

$$\mathcal{H} = \frac{1}{K}\sum_{t=1}^{K}\sum_{j} p_{t,j} \log p_{t,j} = -\frac{1}{K}\sum_{t=1}^{K}\left(\sum_{j=1}^{N_0} p_{t,\text{proper},j} \log p_{t,\text{proper},j} + \sum_{j=1}^{N_1} p_{t,\text{improper},j} \log p_{t,\text{improper},j}\right).$$

Here, $K$ is the total number of decoding steps considered.

## B.2. Experiment

### B.2.1. MODEL HYPERPARAMETER

The stochastic process model has the following hyperparameters:

- The numbers of proper and improper tokens: $N_0, N_1$;

- The logits of proper and improper tokens: $l_{0,\{1,...,N_0\}}, l_{1,\{1,...,N_1\}}$, where:

$$l_{0,i} \sim \mathcal{N}(L_0, \sigma_0), \quad l_{1,i} = L_1;$$

- The number of total steps $K$;

- The noise tolerance rate $\alpha$.

The input is the temperature $T$ and the output is the average token-level entropy $\mathcal{H}$ over 500 seeds. In our experiment, the hyperparameters are set to be:

$$N_0 = 10, \quad N_1 = 30000, \quad L_0 = 0, \quad L_1 = -10, \quad \sigma_0 = 1, \quad K = 512.$$

Furthermore, we tested different noise tolerance rates $\alpha \in [1.5, 2.0, 2.5, 3.0]$ to show the behavior under different noise tolerances.

### B.2.2. RESULT

Figure 8 is the temperature entropy curve derived from the toy model at different noise tolerance rates $\alpha$. The curves with different noise tolerances have a similar shape. Generally, the curve can be divided into two parts: **(sub)-linear** increase and **(super)-exponential** increase. In the first part, the increase is due to the various choices among the proper tokens, while the sharp rise in the second part is due to the loss of control (*i.e.*, the model frequently chooses improper tokens and then makes the error rate extremely high).

In particular, the curve is very similar to the behavior of real language models, and some reference entropy curves and log-entropy curves are shown in Figures 10 and 11.

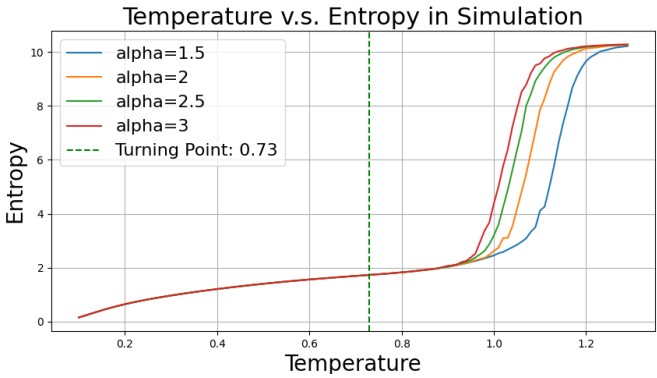

*Figure 6.* The temperature-entropy curves.

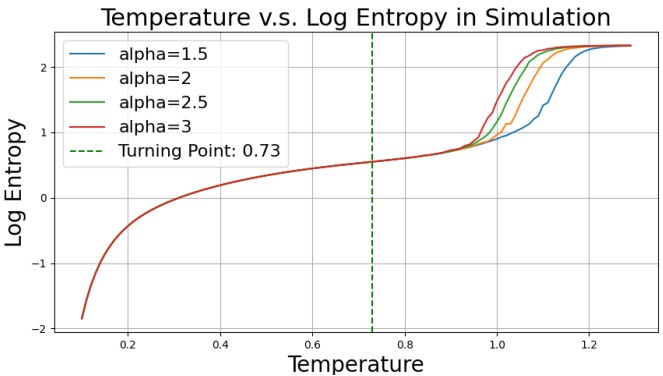

*Figure 7.* The temperature-log entropy curves.

*Figure 8.* The curves derived from the stochastic process model under different $\alpha$.

**Relation to Improper Token Rate** It is natural to consider that proper tokens can lead to correct final answers and that improper tokens will result in incorrect final answers, so we measure the percentage of improper tokens. As shown in Figure 9, when the temperature exceeds the turning point, the percentage of improper tokens increases quickly, implying a quality drop in the samples. Interestingly, the difference in noise tolerance rates has little inference on the turning point but controls the improper token increasing speed after the turning point. However, the percentage of improper tokens increases rapidly in all tested $\alpha$.

### B.3. Conclusion

Our stochastic process model provides a simplified but insightful framework for understanding how temperature-dependent sampling dynamics can lead to characteristic shifts in the model output distribution. The model predominantly chooses from the proper tokens in the low-temperature (or sublinear growth) regime, resulting in relatively stable and controlled outputs. The distribution flattens, and nonsense tokens gain significant probability mass as temperature increases beyond a certain threshold. This transition leads to a sudden and steep increase in entropy—mirroring observations in actual language models—and a corresponding drop in the correct rate. Therefore, increasing the temperature can initially increase generation diversity (sampling among proper tokens) with a small correctness drop. However, the performance suffers a quick drop after reaching a certain threshold (*i.e.*, the turning point EntP).

## C. Aggregation Adaptation Calculation

The choice of aggregation function affects the optimal generation temperature. For example, in majority voting, the final answer must be selected by the majority, whereas in best-of-N, only a single correct instance out of the N samples is required.

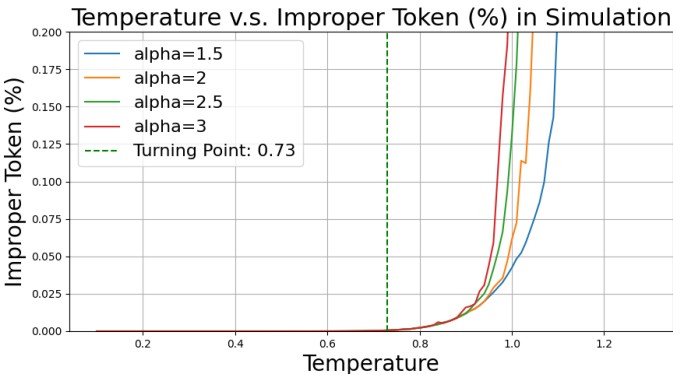

*Figure 9.* The Temperature-Improper Token (%) curves.

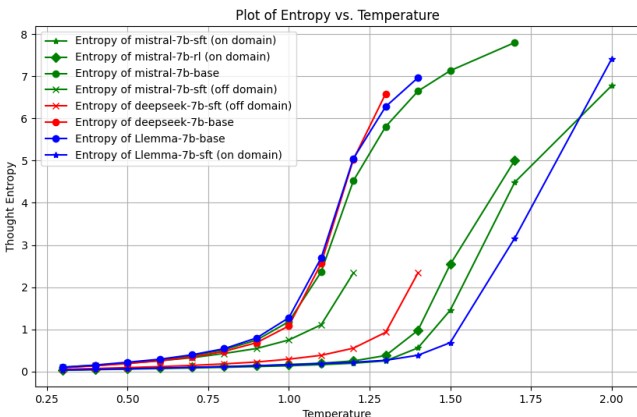

*Figure 10.* The temperature-entropy curves.

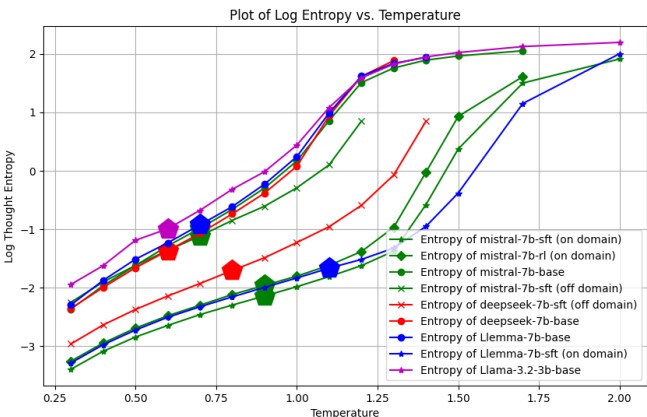

*Figure 11.* The temperature-log entropy curves.

In the case of majority voting, the turning point on the entropy curve aligns with the optimal temperature, so we set its adaptation to 0. For best-of-N, we computed an adaptation on MATH and then tested it on MBPP to confirm generality. Specifically, we averaged the difference between the midpoints of the optimal temperature ranges for best-of-N and majority voting across 13 models on MATH. This difference was $0.092$ on average. Therefore, for simplicity, we set the aggregation adaptation for best-of-N to $0.1$.

*Table 4.* **Aggregation Adaptation for Best-of-N**: we calculate midpoints of optimal temperature ranges on Majority Voting and Best-of-N for MATH. The difference between the average of midpoints is 0.092, so we set the adaptation factor to 0.1.

| Aggregation | Individual Models (Models are listed in the same order as Table 1) | | | | | | | | | | | | | Average |
|---|---|---|---|---|---|---|---|---|---|---|---|---|---|---|
| Best-of-N | 0.6 | 0.8 | 0.6 | 0.6 | 0.7 | 0.5 | 0.6 | 1.1 | 1.2 | 0.5 | 0.6 | 1.3 | 1.0 | 0.7769 |
| Majority Voting | 0.6 | 0.9 | 0.6 | 0.5 | 0.3 | 0.6 | 0.5 | 1.1 | 0.9 | 0.5 | 0.6 | 1.0 | 0.8 | 0.6846 |

## D. Results of All Tested Models

We present the accuracy heatmaps and entropy estimations for all tested models. Figure 12 shows the heatmaps of model accuracy for the MATH dataset, while Figure 13 displays the heatmaps for the MBPP dataset. Additionally, Figure 14 illustrates the entropy curve estimations for the MATH dataset, and Figure 15 provides the entropy curve estimations for the MBPP dataset.

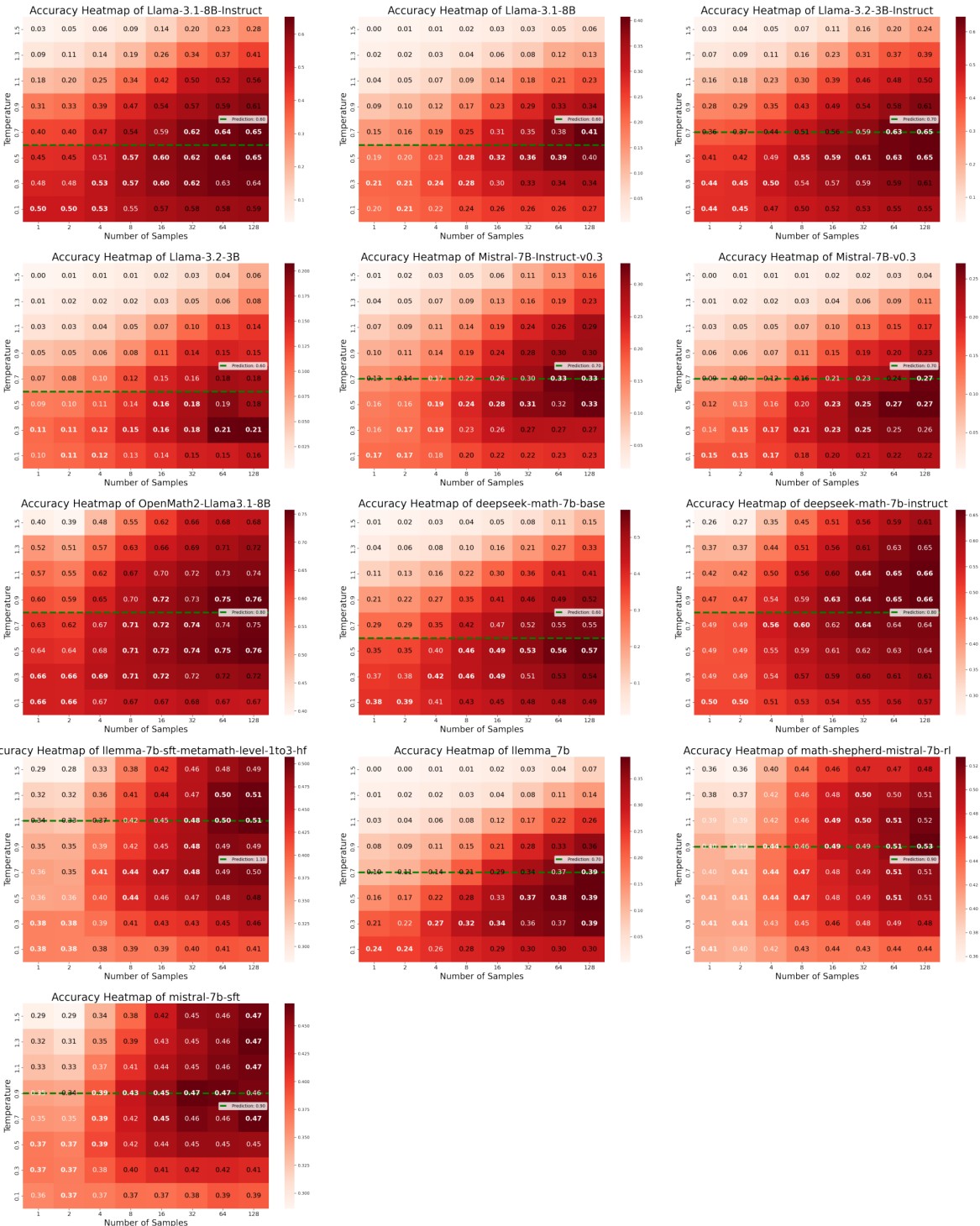

*Figure 12.* The accuracy heatmap for all tested models on the MATH dataset. The green line is our predicted temperature.

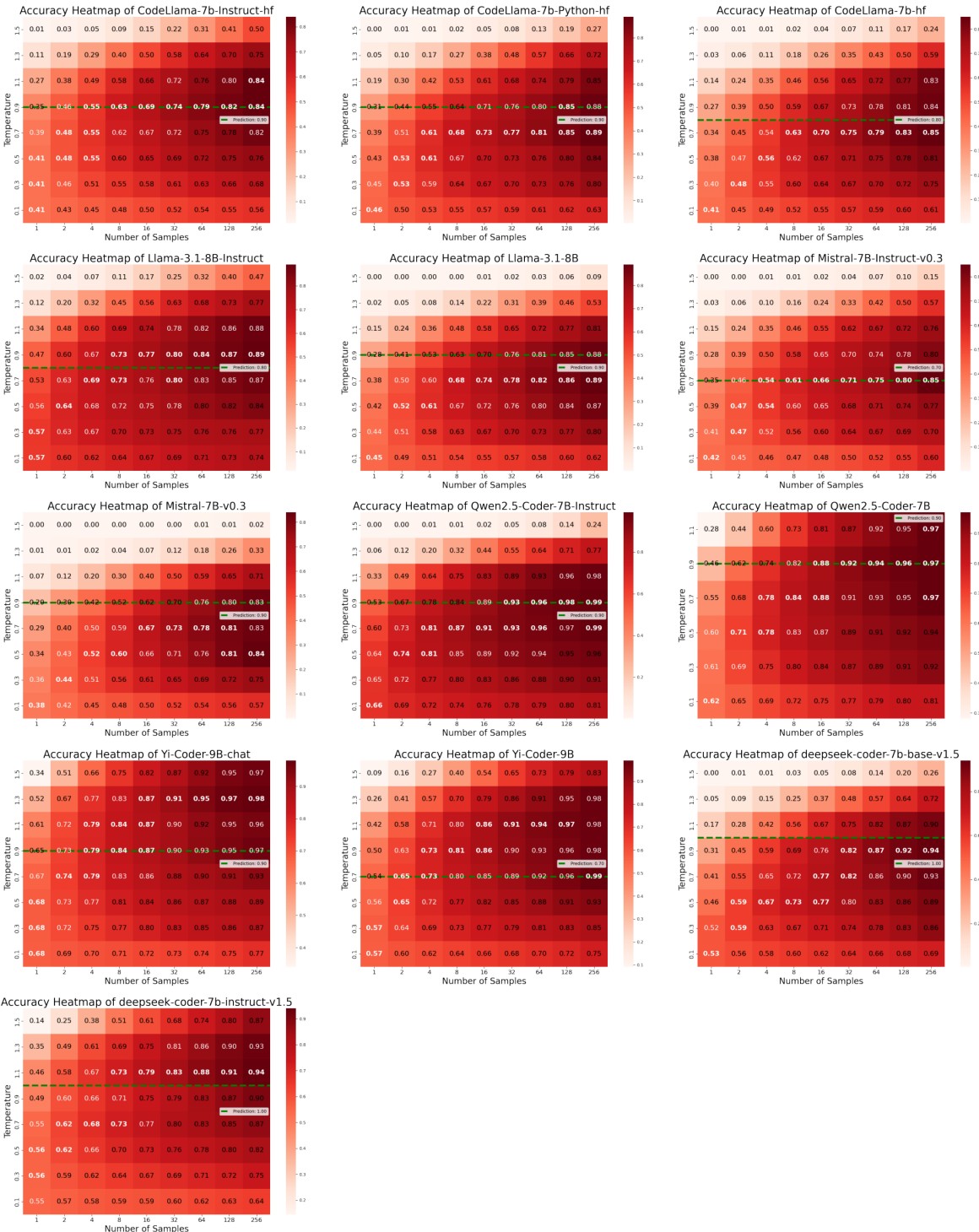

*Figure 13.* The accuracy heatmap for all tested models on the MBPP dataset. The green line is our predicted temperature.

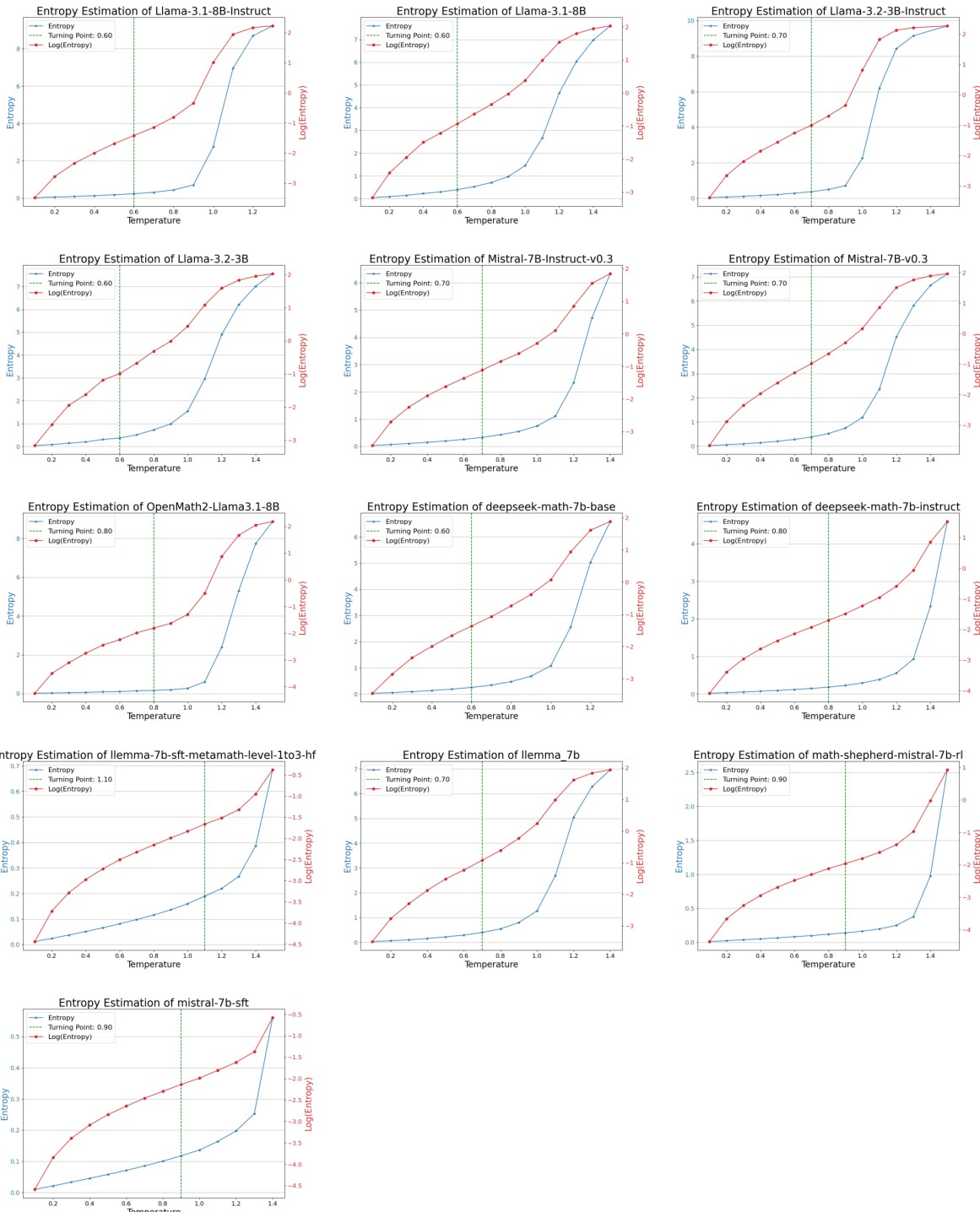

*Figure 14.* The entropy curves and turning points of language models when testing on the MATH dataset.

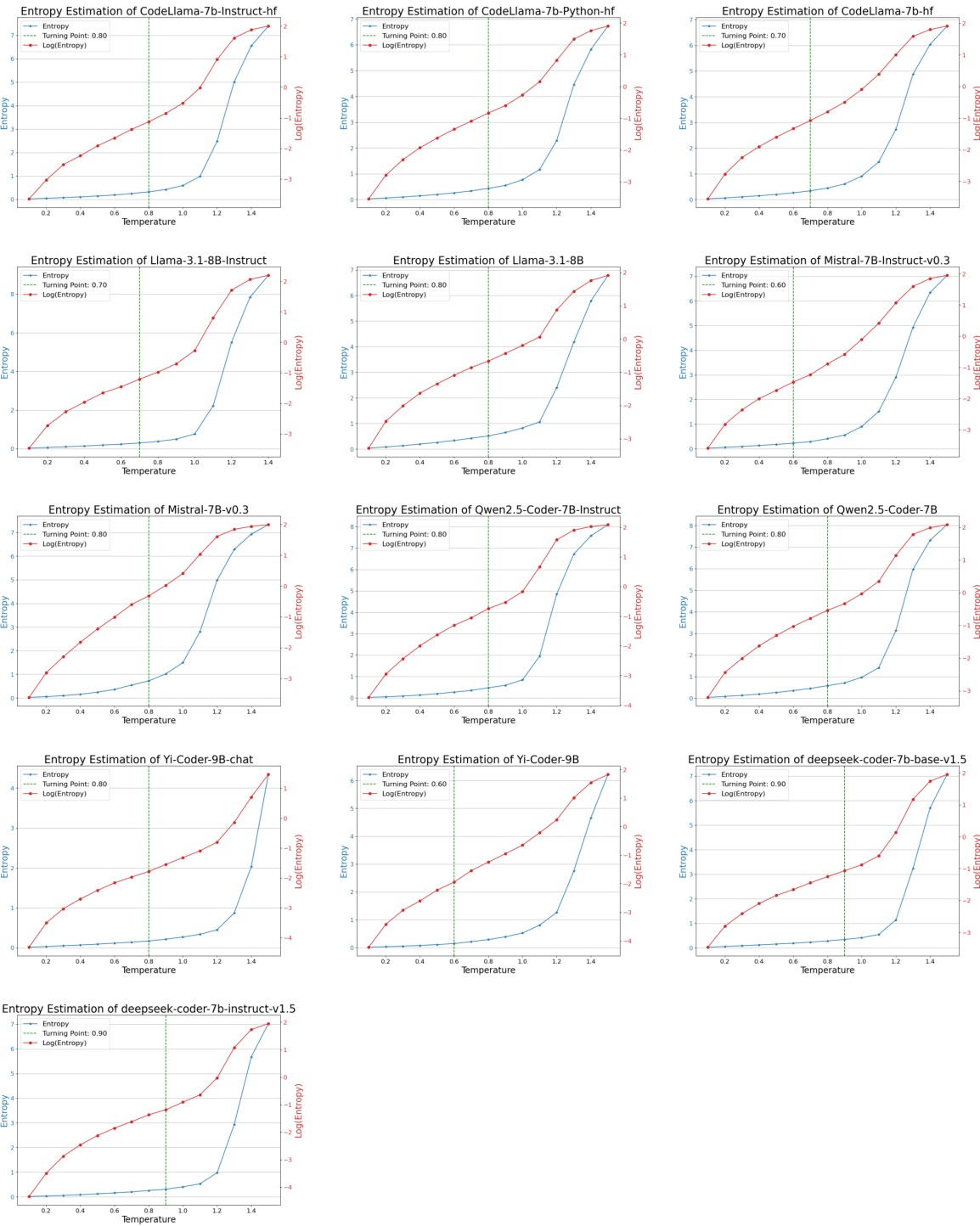

*Figure 15.* The entropy curves and turning points of language models when testing on the MBPP dataset.

