# OpenReview forum: "Optimizing Temperature for Language Models with Multi-Sample Inference"
_ICML.cc/2025/Conference — ICML 2025 poster_

### Official Review · Reviewer_tFFz · 2025-03-14

**Overall Recommendation:** 3

**Summary:**

The authors introduce a data-free way to automatically find the best sampling temperature for multi-sample generation in LLMs. By detecting a sharp “turning point” in the model’s token-level entropy, they find a good balance between quality and diversity.-.

**Claims And Evidence:**

Yes.

**Essential References Not Discussed:**

No.

**Experimental Designs Or Analyses:**

Yes.

**Methods And Evaluation Criteria:**

Yes.

**Other Comments Or Suggestions:**

See Questions.

**Other Strengths And Weaknesses:**

#### Strengths
- The paper is clearly written and well presented.
- It compares multiple LLMs, identifying critical turning points for each. Additionally, it demonstrates that the same LLM can behave differently depending on whether it has been specialized for a given task.

#### Weaknesses
See Questions.

**Questions For Authors:**

1. It is not entirely clear why the authors do not include a comparison to a baseline that has access to labels. Although I understand that TURN does not require labels, selecting the optimal temperature typically occurs before deployment, where labels are commonly available. Could you provide a rationale for excluding this baseline? Additionally, could you elaborate on scenarios where selecting the optimal temperature is necessary but labels are not accessible?

2. Do reasoning models follow the same paradigm? It would be helpful if you could provide results using available reasoning models (e.g., Gemma-3, Qwen).

**Relation To Broader Scientific Literature:**

Research on LLMs is extensive; understanding the impact of temperature is a significant contribution to the field.

**Theoretical Claims:**

There are not theoretical claims.

---

> ### Author Rebuttal · Authors · 2025-03-30
>
> Dear Reviewer tFFz,
>
> Thank you for your thoughtful feedback and for recognizing the significance of our work. We appreciate your positive remarks on our approach and presentation. Our detailed responses are as follows:
>
> > *It is not entirely clear why the authors do not include a comparison to a baseline that has access to labels. Although I understand that TURN does not require labels, selecting the optimal temperature typically occurs before deployment, where labels are commonly available. Could you provide a rationale for excluding this baseline? Additionally, could you elaborate on scenarios where selecting the optimal temperature is necessary but labels are not accessible?*
> >
>
> We compared TURN to grid search using the test set as validation, which serves as a strong upper-bound baseline that has access to labels. Our results show that TURN correlates very highly with grid search (Figure 1B) and even outperforms the best fixed-temperature baseline (Table 2).
>
> In real-world scenarios, labeled data is often scarce or expensive to obtain. For example, in domains like robotics or drug discovery, labeled samples are difficult to collect. Furthermore, when training models that go beyond human performance or require super alignment (e.g., for scientific discovery or working with new or unsolved math problems), ground-truth labels may no longer exist, making label-free methods essential.
>
> > *Do reasoning models follow the same paradigm? It would be helpful if you could provide results using available reasoning models (e.g., Gemma-3, Qwen).*
> >
>
> Thank you for your insightful question about whether reasoning models follow the same paradigm. Evaluating temperature sensitivity in reasoning models is indeed a valuable consideration. At the time of our original submission, the first well-known open-source reasoning model, DeepSeek-R1, was released on January 20, 2025—approximately 10 days before the ICML deadline. This timing prevented us from including such an evaluation in our initial work.
>
> Since then, we have conducted an additional experiment using **DeepSeek-R1-Distill-Qwen-7B** on the MATH dataset. Below are the results (content length = 1000, sample size = 128, majority voting):
>
> | Temperature | 0.1 | 0.3 | 0.5 | 0.7 | 0.9 | 1.1 | 1.3 | 1.5 |
> | --- | --- | --- | --- | --- | --- | --- | --- | --- |
> | Accuracy | 0.847 | 0.860 | 0.855 | **0.874** | 0.860 | 0.830 | 0.830 | 0.800 |
>
> Our findings suggest that temperature has a relatively low impact on the accuracy of **DeepSeek-R1-Distill-Qwen-7B** on the MATH dataset, with performance remaining consistently high across the tested range (peaking at 0.874 at T=0.7). This stability could be attributed to the model being heavily optimized for the MATH dataset and then overfitting it, or the dataset is relatively simple for reasoning models. Due to time constraints, we are unable to explore additional reasoning models or datasets, but we recognize this as a promising direction for future work.
>
> We appreciate your thoughtful review and welcome any further questions or suggestions.

---

> > ### Comment · Reviewer_tFFz · 2025-04-03
> >
> > Thank you for providing additional results on reasoning on MATH (DeepSeek R1). While a more in-depth study would indeed have been beneficial, I find the current work valuable overall. Therefore, I am maintaining my recommendation for acceptance.

---

### Official Review · Reviewer_JAaC · 2025-03-14

**Overall Recommendation:** 3

**Summary:**

Authors explores how to automatically determine the optimal temperature for large language models (LLMs) in multi-sample inference settings, without relying on labeled validation data. The authors analyze temperature’s role in balancing diversity and quality in generated samples and propose TURN (Turning Point Temperature Selection), an entropy-based method for automatic temperature tuning. A key insight is the Entropy Turning Point (EntP)—the temperature at which the log-entropy curve shifts from concave to convex—which strongly correlates with the best-performing temperature found via grid search. Through extensive experiments across diverse models and tasks (e.g., mathematical reasoning and code generation), TURN demonstrates robust generalization and outperforms fixed-temperature baselines, offering a practical solution for optimizing LLM inference.

**Claims And Evidence:**

TURN estimates optimal temperatures without relying on external labels by using token-level entropy. This is a clear advantage over existing methods that require validation data. Also empirical correlations (Fig. 4) between EntP and grid-searched optimal temperatures, showing strong alignment. However, while the results are consistent across datasets, the explanation of why EntP works theoretically is based on a stochastic process model, which is a simplification of real LLM behavior.

**Essential References Not Discussed:**

Not strictly related but also a multi-sample or more specifically multi-domain scaling approach: Robust Calibration with Multi-domain Temperature Scaling. Yaodong Yu, Stephen Bates, Yi Ma, Michael I. Jordan.

**Experimental Designs Or Analyses:**

observed correlation between Entropy Turning Points (EntP) and optimal temperatures supports the method’s validity. The stochastic process model is an interesting theoretical tool to explain entropy behavior, but it remains a simplified approximation that may not capture all aspects of LLM sampling dynamics. The sample efficiency analysis (Table 3) is a strong aspect, demonstrating that TURN requires relatively few samples to make accurate predictions.

**Methods And Evaluation Criteria:**

The method leverages token-level entropy to estimate an optimal temperature without requiring labeled validation data, which is a practical and scalable approach. The evaluation is conducted on two benchmark datasets (MATH for reasoning and MBPP for code generation), which are reasonable choices since they rely heavily on multi-sample aggregation strategies like majority voting and best-of-N. The experiments systematically test different models, including general-purpose and task-finetuned variants, to assess robustness. Key metrics—hit rate, temperature gap, and performance drop—provide a clear measure of how well TURN approximates the best temperature compared to grid search.

**Other Comments Or Suggestions:**

Please include more takaways into fig.1 legend as the concept of EntP is introduced late.

**Other Strengths And Weaknesses:**

I am concerned about the novelty and evaluation here because temperature scaling is common in calibration. Now the paper only measures the entropy turning point and accuracy, it is unclear whether this can apply to canonical setting in calibration - optimizing for a temperature using a holdout validation set to see how the model is better calibrated for decoding. The evaluation does not include a wider range of metrics such as fluency, diversity etc as in the adaptive calibration paper. It is crucial because having a small temperature would
Another concern is it is not common that tuning temperature won't affect model performance a lot if the model size is large enough. [1]

[2] The Effect of Sampling Temperature on Problem Solving in Large Language Models. Matthew Renze, Erhan Guven

**Questions For Authors:**

Usually for math and coding, a good temperature is low, e.g. 0.5, so how does the found optimal temperature vary across different tasks?

**Relation To Broader Scientific Literature:**

The paper's key contributions align with broader research on temperature calibration in language models, multi-sample inference strategies, and self-assessment metrics for optimization. Prior work has established that temperature tuning significantly impacts generation quality, affecting the trade-off between diversity and coherence (Holtzman et al., 2019; Renze & Guven, 2024)

**Theoretical Claims:**

There is no theorem that is proved.

---

> ### Author Rebuttal · Authors · 2025-03-31
>
> Dear Reviewer JAaC,
>
> Thank you for acknowledging our efforts. Our detailed responses are as follows:
>
> > ***Essential References Not Discussed:** Not strictly related but also a multi-sample or more specifically multi-domain scaling approach: Robust Calibration with Multi-domain Temperature Scaling. Yaodong Yu, Stephen Bates, Yi Ma, Michael I. Jordan.*
> >
>
> Thank you for pointing out this reference. *Yu et al.* present a method for training a temperature prediction model with labeled data applicable across multiple tasks for confidence calibration. We will cite and discuss it in our paper. The key distinction is that our study focuses on **accuracy improvement in multi-sample inference** using **label-free temperature selection** in **natural language processing**, whereas the referenced work centers on **confidence calibration** in **computer vision**.
>
> > *I am concerned about the novelty and evaluation here because temperature scaling is common in calibration. Now the paper only measures the entropy turning point and accuracy, it is unclear whether this can apply to canonical setting in calibration - optimizing for a temperature using a holdout validation set to see how the model is better calibrated for decoding.*
> >
>
> While temperature tuning is a well-studied technique in the context of calibration, the novelty of our approach lies in its methodology. As noted by other reviewers (3z2D, NxtV, TgFe, and 3FXt), our method, TURN, determines temperatures based on the **intrinsic probability distribution, without requiring a validation set**. In contrast, traditional temperature calibration relies on **validation data**.
>
> It is also worth highlighting that we include **grid search on test sets** as a baseline in our evaluation. This serves as an **upper bound** for achievable performance. TURN exhibits a **high correlation** between its predicted temperatures and those identified via grid search (Figure 1B), and it **outperforms fixed-temperature baselines** (Table 2).
>
> > *The evaluation does not include a wider range of metrics such as fluency, diversity etc as in the adaptive calibration paper. It is crucial because having a small temperature would*
> >
>
> Our evaluation is focused on **multi-sample inference** strategies like *best-of-N* and *majority voting*, primarily in domains such as math and code, where **accuracy** is an appropriate metric. These tasks are less suited for evaluating fluency or diversity, which are more relevant in open-ended generative tasks.
>
> As for your reference to an "*adaptive calibration*" paper, though without a specific citation, we identified the relevant works:
>
> *[1] Xie et al. Calibrating Language Models with Adaptive Temperature Scaling*
>
> *[2] Huang et al. Calibrating Long-form Generations from Large Language Models*
>
> However, we did not find fluency or diversity as metrics in them. If a different work was intended, we would appreciate a specific reference so we can address it more thoroughly.
>
> > *Another concern is it is not common that tuning temperature won't affect model performance a lot if the model size is large enough. [1]*
> >
> >
> > *[2] The Effect of Sampling Temperature on Problem Solving in Large Language Models. Matthew Renze, Erhan Guven*
> >
>
> Thank you for raising this point. We add an evaluation of a relatively large model (Qwen2.5-32B-Instruct) on MATH at two temperatures using majority voting (sample size = 16):
>
> | Temperature | 0.1 | 0.7 |
> | --- | --- | --- |
> | Accuracy | 0.825 | **0.870** |
>
> Notably, increasing the temperature from 0.1 to 0.7 improved performance by 4.5%, a significant margin given the already high baseline.
>
> While *Renze & Guven* focus on **single-output accuracy**, our study evaluates **multi-sample inference**, where **sample diversity** and **accuracy** are crucial. Thus, our findings do not contradict their conclusions.
>
> > *Please include more takaways into fig.1 legend as the concept of EntP is introduced late.*
> >
>
> Thank you for the constructive suggestion. We have revised Figure 1 and its caption to better introduce and clarify the EnTP concept. The updated version can be found at https://drive.google.com/file/d/1O9DOpEW0z3GRjID6AQeUHT-Z869Tyrqe/view?usp=sharing
>
> > *Usually for math and coding, a good temperature is low, e.g. 0.5, so how does the found optimal temperature vary across different tasks?*
> >
>
> You make a great point. When generating only a few samples (e.g., 4), lower temperatures (0–0.3) often perform better (see Figures 12 and 13). However, with more samples (e.g., 32), low temperatures lead to repetitive outputs, while higher temperatures promote diversity, which is beneficial in multi-sample settings. Additionally, the optimal temperature depends on the task and aggregation methods. For example, best-of-N favors diversity since only one strong candidate is needed, leading to a higher optimal temperature compared to majority voting.
>
> We are looking forward to hearing your further comments!

---

> > ### Comment · Reviewer_JAaC · 2025-04-02
> >
> > Thank you for adding the math result and the new figure. The rebuttal mostly addressed my concerns. I am not quite convinced by the notion of "intrinsic probability distribution" though because the approach did not modify model weights etc to get actionable insights. I increased my score by one.

---

### Official Review · Reviewer_3FXt · 2025-03-15

**Overall Recommendation:** 4

**Summary:**

This paper introduces TURN, an entropy-based approach for automatically determining optimal sampling temperatures for large language models using multi-sample aggregation strategies. The authors identify that different models require different temperature settings based on their training and observe an "entropy turning point" that strongly correlates with optimal temperature values. Their method outperforms fixed-temperature baselines without requiring labeled validation data.

**Claims And Evidence:**

The paper presents two main claims:
1. The optimal temperature varies across models and correlates with training-task similarity
2. The entropy turning point (EntP) can predict optimal temperature without labeled data

Both claims are supported by extensive experimental evidence across multiple models (13 models on two distinct tasks). The correlation between model-task similarity and optimal temperature is compellingly demonstrated.

**Essential References Not Discussed:**

NaN

**Experimental Designs Or Analyses:**

The experiments are thorough, examining 13 models across two distinct tasks. The temperature range exploration is methodical, and the sample size analysis demonstrates TURN's efficiency. However, the work would benefit from 1) testing on more diverse domains beyond mathematical reasoning and code generation, 2) including newer/different aggregation methods (e.g., weighted majority voting)

**Methods And Evaluation Criteria:**

The methodology is sound and the evaluation criteria appropriate. The authors analyze temperature's impact on model performance using three well-defined metrics: Hit Rate, Temperature Gap, and Performance Drop. The comparison against fixed-temperature baselines is comprehensive and fair. One limitation is the restriction to majority voting and best-of-N strategies. Exploring weighted voting would strengthen the work.

**Other Comments Or Suggestions:**

- Investigate scaling laws of optimal temperature with model size would be great
- Analyze whether the optimal temperature varies by difficulty level within tasks1.

**Other Strengths And Weaknesses:**

Strengths:
- Novel insight connecting entropy characteristics to optimal temperature
- Method requires no labeled validation data
- Consistent performance gains across diverse models

Weaknesses:
- Limited task diversity (only math and code)
- Minimal discussion of computational overhead for entropy calculation

**Questions For Authors:**

1. How does TURN perform on generative tasks with more subjective quality metrics?
2. Have you extended your method to tuning other sampling parameters like top-p

**Relation To Broader Scientific Literature:**

The authors adequately position their work within the literature on sampling strategies for LLMs. However, connections to related fields like ensemble methods in traditional ML could be strengthened.

**Theoretical Claims:**

The stochastic process model provides a theoretical foundation for the observed entropy spike, but lacks rigorous mathematical proof. While intuitive and empirically grounded, the connection between the toy model and actual LLM behavior could be more formally established.

---

> ### Author Rebuttal · Authors · 2025-03-31
>
> Dear Reviewer 3FXt,
>
> We appreciate your insightful recommendations. Our point-by-point responses are detailed below:
>
> > *Minimal discussion of computational overhead for entropy calculation*
> >
>
> Table 3 in our paper presents the computational overhead associated with sampling. The results indicate that as few as 40 samples are sufficient for temperature estimation. For entropy computation, we leverage VLLM, which supports efficient inference and provides token probabilities. Calculating the entropy for 40 samples at a temperature interval for an 8B model costs approximately 2 minutes on an A6000 GPU.
>
> > *Investigate scaling laws of optimal temperature with model size would be great*
> >
>
> Thank you for the insightful suggestion. Due to the limited rebuttal period, we may not be able to complete this analysis, but we acknowledge its importance and leave it for future work.
>
> > *Analyze whether the optimal temperature varies by difficulty level within tasks1.*
> >
>
> Thank you for this valuable suggestion. We extended our analysis to test whether the optimal temperature varies by difficulty level on MATH. The table below reports the accuracy across difficulty levels (Model: Llama3.1-8B-Instruct, sample size: 128):
>
> |  | T=0.1 | T=0.3 | T=0.5 | T=0.7 | T=0.9 | T=1.1 |
> | --- | --- | --- | --- | --- | --- | --- |
> | level 1 | 0.90 | **0.91** | **0.91** | 0.90 | 0.88 | 0.86 |
> | level 2 | 0.70 | **0.76** | **0.76** | **0.76** | **0.76** | 0.70 |
> | level 3 | 0.68 | 0.70 | 0.73 | **0.69** | 0.68 | 0.68 |
> | level 4 | 0.45 | 0.51 | 0.54 | **0.59** | 0.46 | 0.36 |
> | level 5 | 0.21 | 0.31 | 0.27 | **0.32** | 0.28 | 0.21 |
>
> While no consistent trend is observed in the optimal temperature across difficulty levels, temperature selection becomes more critical for harder problems. For example, the performance difference between T=0.1 and the optimal temperature is 1% for level-1 problems, but rises to 11% for level-5 problems. This highlights the increasing importance of temperature tuning in complex scenarios.
>
> > *1) testing on more diverse domains beyond mathematical reasoning and code generation
> Limited task diversity (only math and code)
> How does TURN perform on generative tasks with more subjective quality metrics?*
> >
>
> Thank you for these suggestions. We primarily focus on problem-solving domains like math and code, where answers from language models can be aggregated and verified objectively. These domains have demonstrated the most success with multi-sample inference.
>
> Reward models or human evaluations would be required for generative tasks with subjective metrics. However, these evaluation tools may suffer from reward hacking or be expensive. Therefore, to minimize external factors and focus on the language models themselves, we currently limit our experiments to domains with objective quality metrics.
>
> > *2) including newer/different aggregation methods (e.g., weighted majority voting)*
> >
>
> Thank you for the insightful comment. We initially did not experiment with weighted majority voting for two reasons:
> 1. Introducing a reward model could introduce biases, such as reward hacking.
> 2. If the reward model were perfect (i.e., assigning +1 for a correct answer and 0 for an incorrect one), best-of-N and weighted majority voting would yield identical results.
>
> Additionally, to address this within the given time constraints, we conducted an experiment using the log-scale probability of generation as the reward, minimizing external bias. We then applied weighted majority voting and report the results below (Model: Mistral-7B-SFT, Sample Size: 128, Dataset: MATH).
>
> | Temperature | 0.2 | 0.4 | 0.6 | 0.8 | 0.9 | 1.1 | 1.3 | 1.5 |
> | --- | --- | --- | --- | --- | --- | --- | --- | --- |
> | Majority Voting | 0.408 | 0.450 | 0.465 | 0.473 | 0.463 | 0.465 | 0.470 | 0.465 |
> | Weighted Majority Voting | 0.400 | 0.453 | 0.468 | 0.465 | 0.458 | 0.463 | 0.470 | 0.463 |
>
> Empirically, our results indicate that weighted majority voting, when using a reward model with similar performance as the generator, has minimal impact on performance. The optimal temperature range remains almost the same. Furthermore, since entropy estimation in TURN is independent of aggregation methods, TURN remains applicable in this setting.
>
> Future work could explore whether a stronger reward model would provide meaningful benefits in this context.
>
> > *Have you extended your method to tuning other sampling parameters like top-p*
> >
>
> We appreciate this suggestion. We conducted experiments involving other common sampling parameters (Model: Llama3.1-8B-Instruct, sample size: 128, Dataset: MATH):
>
> |  | 0.5 | 0.7 | 0.9 | 1.1 |
> | --- | --- | --- | --- | --- |
> | None | 0.625 | **0.650** | 0.597 | 0.482 |
> | top-p=0.9 | 0.645 | **0.653** | 0.635 | 0.623 |
> | top-k=20 | 0.650 | **0.653** | 0.613 | 0.563 |
>
> These findings suggest that while alternative sampling parameters can slightly reduce temperature sensitivity, they do not significantly impact the optimal temperature.

---

> > ### Comment · Reviewer_3FXt · 2025-04-02
> >
> > Thank the authors for their effort. Since my questions are adequately addressed, I will keep my rating at 4.

---

### Official Review · Reviewer_TgFe · 2025-03-15

**Overall Recommendation:** 4

**Summary:**

The paper tries to find the optimal temperature on various question answering tasks.

They notice that the optimal temperature is higher for models that are fine-tuned on the task, and lower for more general models.

Motivated by that, they set out to find a way to set the temperature that doesn't require a validation set.

To do that, they look at the entropy of generations.  They notice that as you increase the temperature, there is a point at which entropy explodes.  They hypothesise that this is the point at which performance breaks down, which does seem reasonable.

They therefore set the temperature using this point, which they find by looking at the inflection point of log(entropy).

They show modest performance improvements using their temperature selection scheme.

**Claims And Evidence:**

The claims made are appropriate to the evidence.  They include data such as plots of temperature vs token entropy for Llemma-7b (Fig. 1a is this a typo?), and plots of the best accuracy, as obtained from a grid search, against the accuracy at the turning point (Fig 1b).  Figure 2a analyses performance vs temperature and number of samples, while 2b looks at the midpoint of the optimal temperature range vs the number of samples for Mistral-7B-Instruct.

**Essential References Not Discussed:**

None.

**Experimental Designs Or Analyses:**

The experimental designs are appropriate to the claims made, and thorough, e.g. Table 1 looks at multiple models on the MATH and MBPP datasets.

**Methods And Evaluation Criteria:**

Yes.

**Other Comments Or Suggestions:**

N/A

**Other Strengths And Weaknesses:**

Overall, I think the paper has a number of interesting insights, and should be published.

Some other thoughts:
* It would be good to see performance before/after rather than performance drops (as in Table 1).  These performance drops are quite difficult to interpret.
* The performance drops look relatively modest in the Tables, but very large in Figure 4A.  What's going on here?
* I wonder whether what's going is less "temperature is related to closeness of the task", and more "fine-tuned models can in general tolerate higher temperatures than pre-trained models".  Not sure if you did this, but one way to disambiguate would be to take a model fine-tuned on one task, and look at the optimal temperature for a different task.
* Figure 1A is basically repeated as Figure 4A.  I wasn't able to understand Figure 1 on my first reading, so I'd recommend dropping Figure 1.
* Also, Figure 1A/4A has two different y-axes, which is usually regarded as bad-practice.  But I can see why it makes sense here.
* Perhaps the one weakness of the paper is that they find lots of interesting phenomena, such as the explosion in entropy at a particular temperature.  But there isn't that much analysis of these phenomena.  e.g. is there some easy way to see the generations go "off the rails"?  Is it evident in samples?

**Questions For Authors:**

N/A

**Relation To Broader Scientific Literature:**

Yes.

**Theoretical Claims:**

N/A

---

> ### Author Rebuttal · Authors · 2025-03-30
>
> Dear Reviewer TgFe,
>
> Thank you for your supportive review and thoughtful suggestions. Please find our detailed responses below:
>
> > *It would be good to see performance before/after rather than performance drops (as in Table 1). These performance drops are quite difficult to interpret.*
> >
>
> We appreciate this suggestion. Our goal in presenting performance drops was to highlight how close our estimated temperature comes to the true optimal temperature. The complete results are in **Appendix D.** We will add more explanation about performance drops in our revision.
>
> > *The performance drops look relatively modest in the Tables, but very large in Figure 4A. What's going on here?*
> >
>
> Thank you for pointing this out. We believe there may have been a mix-up in the figure reference—**Figure 4A** is unrelated to performance drops. The figure showing drops in accuracy is **Figure 4B**, which visualizes the sensitivity of model performance to temperature.
>
> To clarify:
>
> - **Table 1** compares the performance of our **estimated temperature** to the **best temperature**, resulting in small performance drops due to the accuracy of our estimation.
> - **Table 2** shows performance drops when using a **fixed temperature** (e.g., T=0.1), which can lead to significant degradation (e.g., up to 15.5%) when the temperature is poorly suited to the task.
> - **Figure 4B** aligns with these findings, illustrating how performance deteriorates when the temperature is too low or too high.
>
> > *I wonder whether what's going is less "temperature is related to closeness of the task", and more "fine-tuned models can in general tolerate higher temperatures than pre-trained models". Not sure if you did this, but one way to disambiguate would be to take a model fine-tuned on one task, and look at the optimal temperature for a different task.*
> >
>
> This is a very insightful observation. Our primary claim is that **the optimal temperature correlates with task closeness**—that is, models perform best with higher temperatures when the test task closely matches their training data. However, we added experiments and found that **task-finetuned models applied to unrelated tasks tend to underperform**, making temperature analysis in these **task-mismatch** settings less meaningful.  For example, *CodeLlama-8B*, fine-tuned on code tasks, achieves only **8% accuracy** on the MATH dataset. In such cases, the model’s poor task fit overshadows any insights from temperature tuning.
>
> > *Figure 1A is basically repeated as Figure 4A. I wasn't able to understand Figure 1 on my first reading, so I'd recommend dropping Figure 1.*
> >
>
> Thank you for this constructive feedback. We agree that **Figure 1A** may be hard to grasp before reading the introduction. While it provides early motivation, we recognize the potential redundancy with **Figure 4A**. In our revision, we have refined Figure 1 with clearer captions and context, which can be found at [link](https://drive.google.com/file/d/1O9DOpEW0z3GRjID6AQeUHT-Z869Tyrqe/view?usp=sharing).
>
>
> > *Perhaps the one weakness of the paper is that they find lots of interesting phenomena, such as the explosion in entropy at a particular temperature. But there isn't that much analysis of these phenomena. e.g. is there some easy way to see the generations go "off the rails"? Is it evident in samples?*
> >
>
> This is an excellent point. When the temperature exceeds a certain threshold, we frequently see sharp **entropy spikes**, which come from **erratic or degenerate outputs**. For example, consider the following excerpt from a high-temperature generation on a MATH problem:
>
> > **Step 1: Step 1:** In order to find the probability that the product of the two numbers is a multiple of 5, we will find the total number of possible outcomes, and the number of outcomes that are a multiple of 5.
> … …
> Step 5: Case 1: The first integer is a 5. In that case, the second integer has a **$1\\in\\frac{6}{6}$** probability to be the second integer. Thus the number of ways this can occur is $6$.
> Step 6: Case 2: The second integer is a 5. In that case, the second integer has a **$1\\in\\frac{6}{6}$** probability to be the first integer. Thus the number of ways this can occur is $6$.
> Step 7: Therefore, the probability to obtain the desired product is **$$1-\\frac{6+6}{36}=\\boxed{\\frac{5}{6}}$$**
> The answer is: \\\\boxed{\\frac{5}{6}}
> **In the provided example, the answer to each step in Question 1 is a numerical value or a mathematical expression that … ...**
> >
>
> In this generation, we observe several breakdowns:
>
> - **Repetitions**, such as “Step 1: Step 1”
> - **Logical and numerical inaccuracies**
> - **Unprompted continuation into irrelevant content**
>
> These symptoms clearly illustrate how poor temperature selection, particularly at extremes, can degrade output quality. They also **complement** our entropy-based analysis qualitatively, reinforcing the idea that high-entropy generations often go “off the rails.”

---

### Official Review · Reviewer_NxtV · 2025-03-17

**Overall Recommendation:** 3

**Summary:**

This paper proposes a method to estimate the right temperature LLM. The method relies on  "multi-sample aggregation strategies" which has the advantage to spare the costly and task-specific validation data. The idea is based on an extensive  analysis of the impact of temperature temperature while varying model architectures, datasets, task types, ...  The outcome is a novel entropy-based metric for automated temperature optimization, which allows the model to outperform the  fixed-temperature baselines.

**Claims And Evidence:**

- The paper is well-structured and easy to follow.
- The problem tackled is relevant and interesting.
- The identification of the concave-to-convex shift in entropy as a function of temperature and its connection to model accuracy is particularly insightful.
- The proposed method enables temperature tuning without requiring labeled validation data.

Overall, the demonstration that the proposed method provides a clear advantage over a fixed temperature baseline. This is the central issue since in practice, when using certain models, a quick online search often yields general temperature recommendations, which tend to fall within the presented optimal range. For instance, Llama models commonly use a temperature of 0.7 (see the technical reports), which generally aligns with the authors' reported optimal range across tasks.
This is why clearer formatting and additional comparisons could help make this point more evident.

**Essential References Not Discussed:**

No

**Experimental Designs Or Analyses:**

See the remark above

**Methods And Evaluation Criteria:**

- The experiments appear somewhat disorganized. See remarks for specific concerns.
- The improvements achieved by the method over a fixed temperature setting are marginal, especially given the additional  computational costs of generating multiple samples. See remarks for further discussion.
- The claim that adaptive beta does not require specific tuning lacks sufficient empirical support, as its generality is  demonstrated on only one additional dataset.
- The evaluation process is computationally expensive, and it remains unclear how many samples are actually needed to  reliably estimate the optimal temperature.

**Other Comments Or Suggestions:**

No

**Other Strengths And Weaknesses:**

No

**Questions For Authors:**

See the remark above

**Relation To Broader Scientific Literature:**

Seems OK

**Theoretical Claims:**

N/A

---

> ### Author Rebuttal · Authors · 2025-03-31
>
> Dear Reviewer NxtV,
>
> Thank you for your valuable and positive feedback. We are pleased to hear that you find our research problem compelling and appreciate the novelty of our proposed algorithm. Please find our detailed responses below:
>
> > *For instance, Llama models commonly use a temperature of 0.7 (see the technical reports), which generally aligns with the authors' reported optimal range across tasks. This is why clearer formatting and additional comparisons could help make this point more evident.*
> >
>
> You're absolutely right that T=0.7 is often used as a default for general-purpose models like LLaMA, and in many cases, it performs reasonably well. However, when examining the optimal temperature across different models and tasks, we observe clear deviations from this default, particularly for task-finetuned and pretraining-stage models.
>
> - **Task-finetuned models** often require **higher temperatures** to encourage greater diversity in generation.
> - **Pretraining-stage models**, by contrast, tend to perform better with **lower temperatures** to reduce deviation and improve focus.
>
> For example:
>
> - On the **MATH** dataset, the optimal temperature for *LLaMA3.2-3B* is **T = 0.3**, which results in a **3% performance gain** over the default T = 0.7.
> - Conversely, the optimal temperature for *DeepSeek-Math-7B-Instruct* is **T = 1.0**, outperforming T = 0.7 by **2%**.
>
> These results highlight the importance of adaptive temperature selection across varying model-task combinations. We’ve improved the formatting in our final submission to make these comparisons clearer and more accessible.
>
> > *The improvements achieved by the method over a fixed temperature setting are marginal*
> >
>
> One of our method's key strengths is its ability to operate **without any labeled validation data**, making it directly applicable to real-world test sets. As Reviewer 3z2D also noted, even modest gains can be valuable, particularly given this level of generality and ease of use.
>
> Our method achieves an **average performance improvement of 0.75%** over the best fixed temperature across various models and tasks. Notably, **individual models** see gains of up to **2–3%**, which can be significant in competitive or high-stakes settings.
>
> > *The claim that adaptive beta does not require specific tuning lacks sufficient empirical support, as its generality is demonstrated on only one additional dataset.*
> >
>
> Thank you for highlighting this concern. To further assess the generality of adaptive beta, we conducted an additional experiment on the GSM8k dataset. Due to the time constraints of the rebuttal period, we evaluated only one model (Llama3.1-8B-Instruct, temperature grid size=0.1).
>
> | GSM8k | Majority Voting | Best-of-N | adaptive beta |
> | --- | --- | --- | --- |
> | Llama3.1-8B-Instruct | 0.9 | 1.0 | 0.1 |
>
> Our results show that the adaptive beta calculated for GSM8k (0.1) is consistent with that of MATH (0.092, reported in Appendix C), providing initial evidence that adaptive beta can generalize to another dataset. However, we acknowledge that its behavior may vary across different aggregation functions, and we leave a more comprehensive investigation for future work.
>
> > *The evaluation process is computationally expensive, and it remains unclear how many samples are actually needed to reliably estimate the optimal temperature.*
> >
>
> We address this concern in Section 5.4 (Table 3) of the paper, where we analyze how sample size affects temperature estimation. Our findings show that the method is **robust even with small sample sizes**. For instance, with only **40 samples**, the estimated optimal temperature shows a **variance of only 0.005** and a **performance drop of only 0.2%**. Meanwhile, calculating the entropy for 40 samples at a temperature interval costs approximately 2 minutes on an A6000 GPU for an 8B model. This makes the method practical and cost-efficient, even in low-resource scenarios.

---

### Official Review · Reviewer_3z2D · 2025-03-21

**Overall Recommendation:** 4

**Summary:**

The paper investigates the role of temperature when using best-of-n and majority-voting inference strategies. The authors discover an intriguing phenomenon coined "The Entropy Turning Point" that can accurately predict the change-point between generating diverse high-quality samples and generating low-quality samples. This phenomenon is used to devise an algorithm for automatic temperature selection. In addition, the authors investigate the optimal temperature for pre-trained, instruction-tuned, and domain-fine-tuned LLMs.

**Claims And Evidence:**

For the most part:
- Regarding the proposed distance between the model's train data and a given task, to support the claim that it is indeed an informative distance measure, it'd be interesting to see both code and math models on each of the Figure 3 plots. A good distance measure should identify that code models are further than instruct models from the math task and vice versa.

**Essential References Not Discussed:**

NA

**Experimental Designs Or Analyses:**

- The improvements with the temperature chosen by TURN are relatively modest in comparison to the empirically popular t=0.7. However, given the simplicity of adopting the method, even modest improvements are welcome.
- Best-of-n and majority voting are typically used with CoT-style prompts to achieve the best performance. Could you please provide additional details regarding the prompts used in the experiments? If CoT was not used, I recommend conducting additional experiments with CoT to validate the method in a more practical setting.

**Methods And Evaluation Criteria:**

Yes

**Other Comments Or Suggestions:**

NA

**Other Strengths And Weaknesses:**

NA

**Questions For Authors:**

Please see my questions regarding CoT-prompting and the proposed distance. I'd be happy to increase the score if these questions are addressed.

**Relation To Broader Scientific Literature:**

As test-time scaling gains popularity, choosing temperature is an important consideration.

**Theoretical Claims:**

NA

---

> ### Author Rebuttal · Authors · 2025-03-30
>
> Dear Reviewer 3z2D,
>
> Thank you for the insightful suggestions. We are glad you find our research problem important and provide positive reviews. Please find our detailed responses below:
>
> > *it'd be interesting to see both code and math models on each of the Figure 3 plots. A good distance measure should identify that code models are further than instruct models from the math task and vice versa.*
> >
>
> To further investigate this, we conducted additional experiments comparing the distances between different model types and tasks. Specifically, we measured:
>
> - The distance between the **code-finetuned model** (*code-llama-7b-hf-instruct*) and the **MATH** task,
> - The distance between the **math-finetuned model** (*OpenMath2-Llama3.1-8b*) and the **MBPP (code)** dataset,
> - And the corresponding distances for a **general-purpose model** (*Llama-3.1-8B-Instruct*) for comparison.
>
> The results, shown in the table below, align with expectations: the code-finetuned model is furthest from the MATH task, while the math-finetuned model is furthest from the MBPP task. The general-purpose model falls in between for both. This result implies the robustness of our distance metric.
>
> | Distance | OpenMath2-Llama3.1-8b (finetuned on math) | Llama-3.1-8B-Instruct (general purpose) | code-llama-7b-hf-instruct (finetuned on code) |
> | --- | --- | --- | --- |
> | MATH | **0.0867** | 0.1847 | 0.2152 |
> | MBPP (code) | 1.0159 | 0.1901 | **0.1477** |
>
> > *Could you please provide additional details regarding the prompts used in the experiments? If CoT was not used, I recommend conducting additional experiments with CoT to validate the method in a more practical setting.*
> >
>
> For the **MATH** task, we **did** use chain-of-thought (CoT) prompting. Specifically, we adopted prompts from the official codebase for models already finetuned on the MATH training set with typical CoT reasoning formats (e.g., *llama-7b-sft-metamath-hf*). For general-purpose instruction-tuned models, we added a **four-shot CoT prompt** containing step-by-step reasoning examples.
>
> For the **coding task**, models were prompted to directly generate code solutions, following the default configuration from the *bigcode-evaluation-harness* benchmark.
>
> We hope these clarifications and additional experiments address your concerns. Thank you again for your thoughtful feedback — we welcome any further questions or suggestions.

---

> > ### Comment · Reviewer_3z2D · 2025-04-04
> >
> > Thank you for the rebuttal. My questions have been addressed and I've updated my score.

---

### Decision · Program_Chairs · 2025-05-01

**Decision:**

Accept (poster)

**Comment:**

This paper proposes to optimize the temperature parameter for decoding from language models in inference scaling applications (using best-of-n or majority voting), The paper observes an intriguing phenomenon: The accuracy vs temperature plot seems to go through a turning point going from concave to convex on a certain temperature. The proposal then is to use that temperature for decoding. Subsequent experiments show that the performance is on par with optimizing temperature on a validation set. The key selling point of the proposal is that it is solely based on the entropy of the generation of the model and there is no need for a validation set to choose the temperature. Besides that the paper makes uncovers intriguing phenomena about the optimal temperature in generative models beyond the default 0.7. Overall, I think the concave / convex hypothesis that the paper identifies is quite interesting, and all reviewer and the AC agree with this fact (which is the basis of the positive scores). However, the AC also identified some issues that need to be addressed for the camera ready to the extent that is possible. Congratulations to the authors!

- The message of this paper is slightly misleading. The authors pitch their method against validation over a temperature set of {0.3, 0.5, 0.7, 0.9, 1.1} which might be too coarse given that TURN uses an interval of size 0.1 and show that it has a smaller gap. This gap would likely evaporate and become unfavorable to TURN if both sets use an interval of size 0.1. I am actually not convinced that when there is a validation set to choose temperature, TURN would be any useful. I think the real usefulness is in cases where no such validation set exists for using the model in the wild for new problems.

- The relative metrics are confusing. I think the temperature selection metrics are not too informative. The main important outcome of the process is actually accuracy. I suggest that the authors simply report the raw numbers (e.g., accuracy) and compare the accuracy of using TURN vs optimal vs grid search on the same interval of 0.1.

- The proposed concave / convex hypothesis is based on a heuristic observation. The fact that the algorithm actually returns a valid outcome suggests that the concave / convex hypothesis is likely valid but I think more investigation on that front is needed on why that happens, what is needed for it to happen, etc. Also in practice, the algorithm is at the mercy of numerical stability of estimating the second order statistics. To this end, the authors propose a stochastic process model in Section 4.3, which I think is half baked and I was not convinced that it captures the dynamics of real language models.